# WarriorMath: Enhancing the Mathematical Ability of Large Language Models with a Defect-aware Framework

## Abstract

Large Language Models (LLMs) excel in solving mathematical problems, yet their performance is often limited by the availability of high-quality, diverse training data. Existing methods focus on augmenting datasets through rephrasing or difficulty progression but overlook the specific failure modes of LLMs. This results in synthetic questions that the model can already solve, providing minimal performance gains. To address this, we propose WarriorMath, a defect-aware framework for mathematical problem solving that integrates both targeted data synthesis and progressive training. In the synthesis stage, we employ multiple expert LLMs in a collaborative process to generate, critique, and refine problems. Questions that base LLMs fail to solve are identified and iteratively improved through expert-level feedback, producing high-quality, defect-aware training data. In the training stage, we introduce a progressive learning framework that iteratively fine-tunes the model using increasingly challenging data tailored to its weaknesses. Experiments on six mathematical benchmarks show that WarriorMath outperforms strong baselines by 12.57% on average, setting a new state-of-the-art. Our results demonstrate the effectiveness of a defect-aware, multi-expert framework for improving mathematical ability.

## 1 Introduction

Large language models (LLMs) have demonstrated remarkable capabilities in solving mathematical and scientific problems (Jaech et al., 2024; Anthropic, 2025; He et al., 2024b; Zeng et al., 2024; OpenAI, 2025; DeepMind, 2025; DeepSeek-AI et al., 2025), positioning them as valuable mathematical assistants. Consequently, enhancing their mathematical ability has become a key research goal. Yang et al. (2024) improve math skills through large-scale pre-training on math data, while Muennighoff et al. (2025); Wen et al. (2025); Min et al. (2024) focus on fine-tuning with high-quality instruction datasets. However, both approaches heavily rely on high-quality data (Xu et al., 2024; Feng et al., 2025), and key training data for strong models (e.g., OpenAI o1 (OpenAI, 2025)) remain private, limiting reproducibility. Thus, collecting and annotating high-quality math problems at scale remains a major bottleneck.

In response, recent research has explored large-scale data synthesis using LLMs to augment training datasets (Tang et al., 2024; Huang et al., 2025; Yue et al., 2024; Liu et al., 2024; Zhou et al., 2024; 2025; Li et al., 2024a; He et al., 2025b; Luo et al., 2025a; Li et al., 2024b; Mei et al., 2025). Some of these methods focus on mining instruction data from pretraining corpora (Yue et al., 2023; Li et al., 2024d), while others generate new data by rephrasing existing problems (Yu et al., 2023) or employing difficulty progression techniques (Xu et al., 2024; Luo et al., 2025a).

While these data synthesis approaches have made progress, they are not tailored to address the specific deficiencies of the base LLM. As a result, they predominantly generate easily solvable problems that do not effectively challenge the model, offering limited learning benefits (see Figure 1). *Inherent defects*, such as misinterpretations of quantifiers or incorrect symbolic steps (Pan et al., 2025; An et al., 2024), are often overlooked. Unlike difficulty, defects reflect internal model limitations. By ignoring these defects, existing synthesis methods primarily generate problems that the model can already solve, hindering further improvement (Yu et al., 2025; Pan et al., 2025; An et al., 2024).

(a) Existing data synthesis strategy. Seed datasets are collected, and an external LLM is prompted to augment or label them. However, the resulting problems are often too simple, leading to limited improvements in model performance.

(b) Our defect-aware strategy. An Exam Committee of multiple expert LLMs generates, critiques, and refines problems, retaining only the failure data where the base LLMs struggled. This ensures synthesized problems challenge the base LLMs and enhance their capabilities.

Figure 1: Comparisons between our method and traditional data synthesis strategies.

Recent research has shown that aligning training strategies with the model's evolving defects can significantly improve performance on challenging tasks (An et al., 2024; Wen et al., 2025; Yu et al., 2025; Teams, 2024). Therefore, a more interactive and adaptive synthesis process is required to effectively address these defects.

Based on this insight, we introduce **WarriorMath**, a defect-aware framework for mathematical problem solving that combines both data synthesis and progressive training. As shown in Figure 2, WarriorMath decomposes data synthesis into two stages: **(1) Defect-aware problem construction.** We assemble an exam committee of state-of-the-art mathematical LLMs, each contributing its expertise to generate challenging problems. These problems are then reviewed by judges, while the base LLM provides feedback, identifying the specific areas it has yet to master. **(2) Answer generation and refinement.** Each model attempts to solve the generated problems, and the solutions are filtered and ranked based on Elo ratings and voting. This ensures that the solutions produced are not only correct but also provide meaningful guidance for the base LLM's learning. Unlike previous methods that expand existing datasets, WarriorMath generates novel, defect-specific training examples from scratch, allowing for more efficient and targeted model improvement.

For training, we introduce a **progressive learning** framework that systematically addresses the model's defects through a two-stage process. WarriorMath begins with supervised fine-tuning (SFT) using answers generated by multiple expert models, thus establishing a broad foundation of mathematical knowledge. We then identify the failure problems that the model still struggles with and fine-tune it to prioritize stronger solutions on these problematic examples. This process is repeated iteratively, allowing the model to progressively correct its inherent defects without overwriting previously mastered concepts, thereby facilitating more effective learning from its mistakes.

To evaluate the effectiveness of WarriorMath, we conduct evaluations on six prevalent mathematical benchmarks (AIME-2024, 2024; AIME-2025, 2025; AMC-2023, 2023; Lightman et al., 2024; Lewkowycz et al., 2022; He et al., 2024a). Evaluation on these benchmarks indicates that WarriorMath achieves SOTA performance, surpassing existing same-sized open-source large models by an average of 12.57%. Notably, the ablation experiments demonstrate that the proposed synthesis strategy can indeed generate proper high-quality data for the base LLM, as well as the effectiveness of the proposed training framework in learning from defects.

The key contributions of this work include:

- We propose a defect-aware data synthesis pipline, which generate proper and high-quality data designed for base LLMs from scratch which emulates the educational philosophy of teaching according to aptitude.

- We introduce a progressive learning framework that first learns broadly from experts and then improve ability through iterative alignment focusing on reinforcing knowledge where the model fails while bypassing mastered knowledge.

- We demonstrate that **WarriorMath** achieves state-of-the-art performance among open-source LLMs, with strong data efficiency and generalization, validating the effectiveness of our approach.

## 2 Related Work

### 2.1 Math LLMs

Recent advances in LLMs' mathematical capabilities have drawn growing attention from both academic and industrial communities. Early successes were fueled by the creation of large-scale pretraining corpora and curated fine-tuning datasets (Paster et al., 2023; Wang et al., 2024; Shao et al., 2024; Yue et al., 2023), which significantly improved model accuracy on standard math benchmarks. This progress was further accelerated by specialized prompting strategies (Wei et al., 2022; Imani et al., 2023), tool augmentation (Gao et al., 2023; Schick et al., 2024), and reinforcement learning techniques (DeepSeek-AI et al., 2025; Zhao et al., 2024). While advanced prompting and test-time scaling methods (Wu et al., 2024; Muennighoff et al., 2025) continue to push performance limits, current LLMs still lag behind those of proprietary ones (e.g. GPT-4o, Claude, etc.), primarily because stronger models often keep their training data proprietary (Hui et al., 2024). As a result, the lack of publicly available high-quality, diverse datasets remains a significant barrier to further development in this field.

### 2.2 Data Synthesis

Synthetic data has been employed to augment training datasets for various mathematical LLMs (Luo et al., 2025a; Teams, 2024; Li et al., 2024a). Early approaches follow the Self-Instruct paradigm (Wang et al., 2023), using few-shot prompting to generate synthetic instructions. To enhance diversity and difficulty, recent work (Luo et al., 2025a; An et al., 2024; Liu et al., 2024) further explores iterative refinement and instruction evolution based on reasoning trajectories. While these methods improve the quality and diversity of synthetic data, they primarily focus on difficulty progression rather than addressing the model's actual capability defects. That is, increasing difficulty does not necessarily target the failure cases of base LLMs. As a result, many synthesized questions remain solvable by the model, providing limited value for improving model ability. Moreover, many of these approaches depend heavily on proprietary LLMs (e.g., GPT-4, Claude) (Muennighoff et al., 2025), making large-scale data generation costly and less reproducible. Recent work such as Feng et al. (2025) proposes a novel paradigm that distills data through multi-agent competitions among open LLMs, reducing reliance on external APIs. However, these methods still overlook the importance of targeting model-specific defects during synthesis.

### 2.3 Learning From Defect

An emerging line of work investigates how failure cases can be leveraged to guide LLMs toward improved performance. Reflexion (Shinn et al., 2023) introduces a self-reflection mechanism in which the model analyzes past failures using either internal reflections or external feedback. Similarly, Gou et al. (2024) use external tools to provide real-time critiques, while Chen et al. (2024) enable models to execute and debug code to enhance factual consistency. In contrast to these tool-based feedback approaches, WizardLM-2 (Teams, 2024) proposes the AI-Align-AI (AAA) framework, where multiple LLMs collaboratively teach and critique each other. This setup involves simulated dialogues, quality evaluations, and constructive suggestions for improvement, allowing models to iteratively refine their outputs in a multi-turn process. Despite these advances, most methods still treat failure feedback as a post-hoc augmentation rather than an integral part of the training process. Furthermore, few approaches systematically incorporate failure-driven supervision into dataset synthesis pipelines, leaving a gap between training data construction and the model's evolving weaknesses.

## 3 Method

In this section, we elaborate on the details of our WarriorMath. As illustrated in Figure 2, the pipeline mainly contains two components: Defect-aware data synthesis and Progressive training. The details of data synthesis will be presented in § 3.1 and the training method will be described in § 3.2

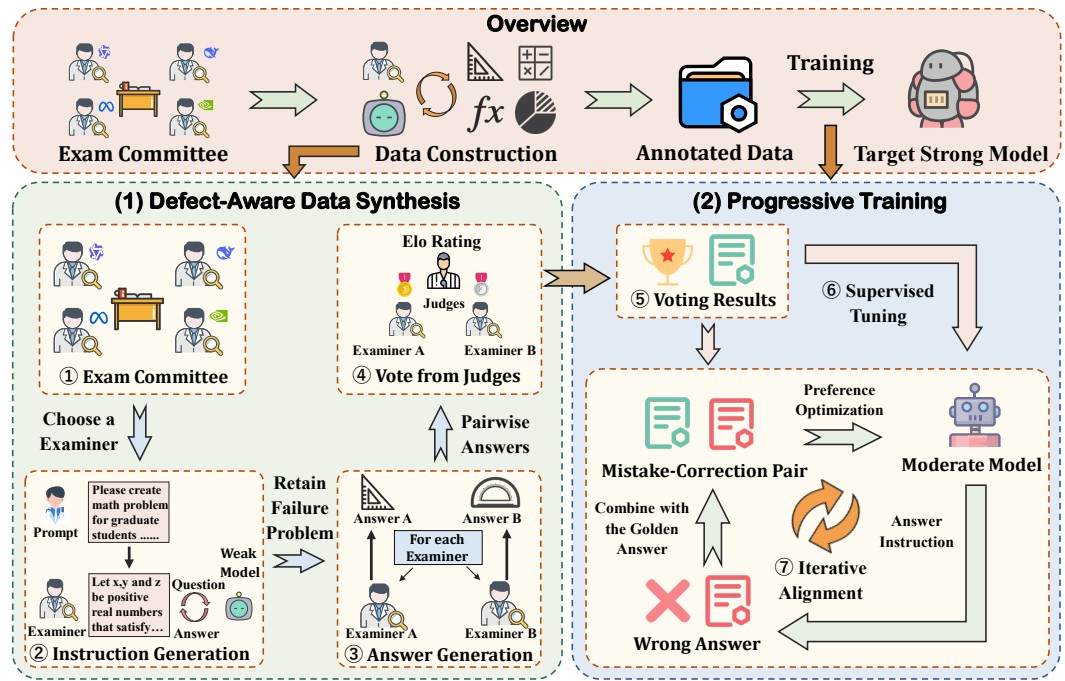

Figure 2: **The WarriorMath's pipeline.** The WarriorMath pipeline includes two key components: **Defect-Aware data synthesis** and **Progressive Training**. (1) A multi-expert committee generates problems via the Examiner. The Base Model identifies its own weaknesses by attempting these problems, retaining only the failed problems. Solvers generate competing solutions, and Judges use Elo ratings to select the authoritative Answer, constructing the training data. (2) The model undergoes Supervised Fine-Tuning (SFT). It then enters the Iterative Defect Alignment loop, where the model's failed responses (hard negatives) are paired with the Golden Answers and used for Preference Optimization (DPO-based) to systematically overcome the identified defects.

## 3.1 DEFECT-AWARE DATA SYNTHESIS

Unlike conventional methods that expand existing datasets, WarriorMath generates novel, defect-specific training examples from scratch. We employ a collaborative **Exam Committee** of expert LLMs to generate, solve, and critique mathematical problems. To ensure clarity in our multi-agent framework, we explicitly define three distinct roles for the committee members: the **Examiner** (generates the problem), the **Solver** (proposes solutions), and the **Judge** (evaluates solution quality).

### 3.1.1 PROBLEM GENERATION

**Committee members Setting.** The effectiveness of **WarriorMath** is directly linked to the strength and diversity of its members. For this study, we select five leading open-source math LLMs, DeepSeek-R1-Distill-Llama-70B (DeepSeek-AI et al., 2025), Qwen2.5-Math-72B-Instruct (Yang et al., 2024), QwQ-32B (Qwen, 2025), AceMath-72B-Instruct (Liu et al., 2025b) and Phi-4-reasoning (Abdin et al., 2025). In each synthesis round, one model is selected as the Examiner, while the others act as Solvers or Judges.

**Problem Synthesis from Scratch** The goal is to leverage the Examiner's capabilities to pose challenging questions to expose the defects of the **Base Model**. Unlike previous approaches where problems are generated indiscriminately, our design ensures the Examiner focuses on generating problems that the current Base Model is likely to fail. We design a prompt that instructs the Examiner to act as a world-class expert generating difficult mathematical problems. To achieve this, we design a prompt that instructs LLM A to act as a world-class expert in generating difficult and diverse mathematical problems. Specifically, the prompt emphasizes four dimensions: (1) Quality, requiring

problems to be clear, well-structured, and unambiguous; (2) Difficulty, demanding deep mathematical reasoning beyond pattern matching; (3) Diversity, covering a broad set of domains such as algebra, calculus, discrete mathematics, and geometry; and (4) Challenge, encouraging adversarial elements like subtle traps or misleading intermediate steps. This deliberate prompt design ensures LLM A activates its mathematical capabilities while producing maximally diagnostic questions to evaluate LLM B. Further Problem Synthesis details are provided in Appendix B.

**Deduplication and Defect-Aware assessment.** To ensure the quality and informativeness of selected problems, we identify and filter out problems that are repetitive, ambiguous, or excessively difficult for the base LLM to learn from effectively. The Prompt for selection and discarded problems are provided in appendix B.3. To further ensure that the retained problems are truly valuable for learning, we conduct a defect-aware assessment of the model's performance. For each problem, we generate $N = 16$ rollouts using the base model. After verifying the correctness of these outputs, we discard problems for which all sampled solutions are correct, retaining only those that the model answers incorrectly. Through interactive assessment, we confirm that these retained problems continue to provide meaningful learning signals. Finally, to preserve both the diversity and representativeness of the instruction set, we apply the KCenterGreedy algorithm (Sener & Savarese, 2018) to select a final subset $\bar{\bar{I}}$, using the *all-roberta-large-v1* embedding model (Liu et al., 2019) to compute semantic similarity between instructions.

### 3.1.2 ANSWER GENERATION AND REFINEMENT

Once a defect-aware problem is identified, we should generate a high-quality "Golden Answer." This is achieved through a competitive refinement phase involving Solvers and Judges.

**Answer Generation (The Solvers).** For every instruction $i$, we assign two models to the role of Solver. Solver A (The Examiner): The model that created the problem must provide its own solution. Solver B (Peer Expert): A different committee member provides an independent solution. This generates a pairwise set of answers for evaluation. **Answer Refinement (The Judges)**. The remaining committee members act as Judges, voting on the correctness and helpfulness of the paired responses (more details about can be found in AppendixB.4). We first calculate the *local score* based on the raw vote counts $(t_A, t_B)$

$$x^i_{A>B} = \frac{t_A}{t_A + t_B} \quad x^i_{B>A} = \frac{t_B}{t_A + t_B} \tag{1}$$

Here, $x^i_{A>B}$ represents the percentage of votes Solver A receives, while $x^i_{B>A}$ similarly represents the percentage of votes Solver B receives. $t_A$ and $t_B$ are the raw vote counts for A and B.

**Elo Rating Integration.** Relying solely on local votes can be problematic due to judge bias or stochasticity, potentially allowing weaker models to win unrepresentatively. To address this limitation and enforce **global consistency**, We introduce the concept of *the Elo rating* Bai et al. (2022), which provides a more comprehensive reflection of a model's relative performance over time and across various evaluations.

$$X^{Elo}_{A>B} = \frac{1}{1 + 10^{(R_B - R_A)/400}}$$
$$X^{Elo}_{B>A} = \frac{1}{1 + 10^{(R_A - R_B)/400}} \tag{2}$$

where $X^{Elo}_{A>B}$ and $X^{Elo}_{B>A}$ indicate the expected probabilities of A defeating B and B defeating A, respectively. $R_A$ and $R_B$ are the Elo rating of A and B, which are updated dynamically and iteratively. The update rule is:

$$R_A \leftarrow R_A + K \times (s^i_{A>B} - X^{Elo}_{A>B})$$
$$R_B \leftarrow R_B + K \times (s^i_{B>A} - X^{Elo}_{B>A}) \tag{3}$$

where $s^i$ is the actual outcome (1 for win, 0.5 for draw, 0 for loss) and $K$ controls sensitivity.

Based on Equation 1 and Equation 2, we calculate the definitive score for Solver A's response to instruction $i$ by balancing the local vote with the global Elo expectation:

$$e_A^i = \sum_{B \in Com \setminus A} \alpha X_{A>B}^{Elo} + (1-\alpha)x_{A>B}^i \tag{4}$$

where $Com$ is the set of all the solvers and '$\setminus$' is the subtraction operation. $\alpha$ is the coefficient to balance the local contingency and global consistency.

## 3.2 PROGRESSIVE TRAINING

We introduce a **Progressive Training** framework that incrementally improves a model's mathematical reasoning by systematically identifying and correcting its errors. The framework consists of two stages: supervised fine-tuning (SFT) and iterative alignment.

**Stage 1: Supervised Fine-Tuning.** Given a dataset $\mathcal{D} = \{(x_i, Y_i, \{r_i^j\}_{j=1}^N)\}_{i=1}^N$, where $x_i$ is an instruction, $Y_i = \{y_i^j\}_{j=1}^N$ are expert responses, and $r_i^j$ are their scores, we select the highest-scoring response as $y_i^{\text{gold}} = \arg\max_{y_i^j \in Y_i} r_i^j$. The gold pairs $(x_i, y_i^{\text{gold}})$ are then used to initialize the model $M_0$ through maximum likelihood estimation.

**Stage 2: Iterative Alignment.** To further refine the model, we adopt the iterative alignment strategy of Pang et al. (2024), where the model improves by learning from its own mistakes. At iteration $t$, the current model $M_t$ generates a set of candidate responses

$$G_i = \{(c_i^n, y_i^n)\}_{n=1}^{N_i} \sim M_t(x_i), \tag{5}$$

with reasoning traces $c_i^n$ and final answers $y_i^n$. Each response is labeled for correctness by $r_i^n = \mathcal{R}(y_i^n, \hat{y}_i)$, which reduces to $r_i^n = 1$ if $y_i^n = \hat{y}_i$, and 0 otherwise. This yields a labeled set $G_i = \{(c_i^n, y_i^n, r_i^n)\}$ and the subset of incorrect responses

$$G_i^{\text{neg}} = \{(c_i^n, y_i^n) \mid r_i^n = 0\}. \tag{6}$$

For each gold pair $(c_i^{\text{gold}}, y_i^{\text{gold}})$, we construct preference pairs by contrasting it with negatives $(c_i^l, y_i^l) \in G_i^{\text{neg}}$, forming $D_t^{\text{pairs}}$. The model is optimized with a hybrid objective:

$$\begin{aligned}
\mathcal{L}_{\text{total}} &= \mathcal{L}_{\text{DPO}} + \alpha \mathcal{L}_{\text{NLL}}, \\
\mathcal{L}_{\text{DPO}} &= -\log \sigma \Big( \beta \log \frac{M_\theta(c_i^{\text{gold}}, y_i^{\text{gold}}|x_i)}{M_t(c_i^{\text{gold}}, y_i^{\text{gold}}|x_i)} \\
&\quad - \beta \log \frac{M_\theta(c_i^l, y_i^l|x_i)}{M_t(c_i^l, y_i^l|x_i)} \Big), \\
\mathcal{L}_{\text{NLL}} &= -\frac{1}{|c_i^{\text{gold}}|+|y_i^{\text{gold}}|} \log M_\theta(c_i^{\text{gold}}, y_i^{\text{gold}} \mid x_i).
\end{aligned} \tag{7}$$

Here $\sigma$ is the sigmoid, and $\alpha, \beta$ are hyperparameters. After each optimization step, the updated model $M_{t+1} = M_\theta$ is used to generate new responses for the next iteration.

Through repeated self-correction, the model is progressively aligned with expert reasoning, thereby accumulating the collective expertise of diverse mathematical committee members.

## 4 EXPERIMENT

### 4.1 EXPERIMENTAL SETUP

**Backbones.** We implement WarriorMath with two initialization backbones: (1) WarriorMath-Qwen, initialized from Qwen2.5-Math-7B Yang et al. (2024); (2) WarriorMath-DS, initialized from DeepSeek-R1-Distill-Qwen-7B (DeepSeek-AI et al., 2025). As for the competitors of expert battles, we choose strong open-source LLMs including DeepSeek-R1-Distill-Llama-70B (DeepSeek-AI et al., 2025), Qwen2.5-Math-72B-Instruct (Yang et al., 2024), QwQ-32B (Qwen, 2025), AceMath-72B-Instruct (Liu et al., 2025b), Phi-4-reasoning (Abdin et al., 2025).

Table 1: Evaluation results across six mathematical reasoning benchmarks. We report Pass@1 accuracy (mean ± std) of all models across six math benchmarks under a standardized evaluation setup—results are averaged over ten seeds for AIME and AMC, and three seeds for the rest.

| Models | Base | AIME'24 | AIME'25 | AMC'23 | MATH500 | Minerva | Olympiad |
|---|---|---|---|---|---|---|---|
| *Expert Teacher Models* | | | | | | | |
| Qwen2.5-Math-72B-Instruct (Qwen et al., 2025) | Qwen2.5-Math-72B | 32.0±5.9 | 26.3±7.3 | 59.7±6.1 | 85.2±0.5 | 44.1±2.2 | 49.0±0.5 |
| AceMath-72B-Instruct (Liu et al., 2025b) | Qwen2.5-Math-72B | 31.3±7.7 | 28.8±2.2 | 60.1±2.4 | 86.1±2.3 | 57.0±1.6 | 48.4±1.3 |
| DeepSeek-R1-Distill-Llama-70B (DeepSeek-AI et al., 2025) | Llama-3-70B | 67.0±1.9 | 55.3±5.7 | 96.8±2.1 | 95.1±0.7 | 45.1±1.7 | 73.8±0.5 |
| QwQ-32B (Qwen, 2025) | Qwen2.5-32B | 76.3±3.3 | 69.0±4.5 | 96.2±2.4 | 97.5±0.6 | 49.0±0.2 | 78.1±1.0 |
| Phi-4-reasoning (Abdin et al., 2025) | Phi-4-14B | 74.6±5.1 | 63.1±6.3 | 96.0±2.7 | 97.0±0.3 | 49.8±0.2 | 78.0±0.8 |
| *Qwen-Based Models* | | | | | | | |
| Qwen2.5-Math-7B-Base Yang et al. (2024) | - | 20.7±3.8 | 8.7±3.9 | 56.2±5.7 | 64.3±0.5 | 17.3±1.9 | 29.0±0.5 |
| Qwen2.5-Math-7B-Instruct (Yang et al., 2024) | Qwen2.5-Math-7B | 15.7±3.9 | 10.7±3.8 | 67.0±3.9 | 82.9±0.1 | 35.0±0.6 | 41.3±0.9 |
| Qwen-2.5-Math-7B-SimpleRL-Zoo (Zeng et al., 2025) | Qwen2.5-Math-7B | 22.7±5.2 | 10.7±3.4 | 62.2±3.6 | 76.9±1.8 | 30.1±2.8 | 39.3±0.6 |
| Qwen2.5-Math-7B-Oat-Zero (Liu et al., 2025a) | Qwen2.5-Math-7B | 28.0±3.1 | 8.8±2.5 | 66.2±3.6 | 79.4±0.3 | 34.4±1.4 | 43.8±1.1 |
| LIMR (Li et al., 2025) | Qwen2.5-Math-7B | 30.7±3.2 | 7.8±3.3 | 62.2±3.4 | 76.5±0.4 | 34.9±1.3 | 39.3±0.9 |
| Qwen2.5-7B-Instruct (Qwen et al., 2025) | Qwen2.5-7B | 12.3±3.2 | 7.3±3.4 | 52.8±4.8 | 77.1±1.2 | 34.9±1.0 | 38.7±1.0 |
| s1.1-7B (Muennighoff et al., 2025) | Qwen2.5-7B | 19.0±3.2 | 21.0±5.5 | 59.5±3.7 | 80.8±0.6 | 37.5±1.1 | 48.2±1.4 |
| Eurus-2-7B-PRIME (Cui et al., 2025) | Qwen2.5-7B | 17.8±2.2 | 14.0±1.7 | 63.0±3.9 | 80.1±0.1 | 37.5±1.0 | 43.9±0.3 |
| Bespoke-Stratos-7B (Bespoke Labs, 2024) | Qwen2.5-7B | 20.3±4.3 | 18.0±4.8 | 60.2±4.9 | 84.7±0.5 | 39.1±1.3 | 51.9±1.1 |
| WarriorMath-Qwen-7B | Qwen2.5-7B | 48.3±2.6 | 36.5±5.0 | 83.0±2.5 | 88.3±1.4 | 41.2±2.8 | 52.1±0.8 |
| *DeepSeek-Based Models* | | | | | | | |
| DeepSeek-R1-Distill-Qwen-7B (DeepSeek-AI et al., 2025) | - | 52.3±6.3 | 39.0±5.9 | 91.5±2.7 | 94.1±0.3 | 40.1±0.4 | 67.3±0.1 |
| Light-R1-DS-7B (Wen et al., 2025) | DeepSeek-R1-Distill-Qwen-7B | 53.0±4.8 | 41.0±3.5 | 90.0±3.1 | 93.5±0.5 | 41.3±1.3 | 68.0±1.2 |
| AReaL-boba-RL-7B (inclusionAI, 2025) | DeepSeek-R1-Distill-Qwen-7B | 56.7±9.2 | 40.0±9.1 | 90.0±4.8 | 94.4±1.0 | 40.8±3.0 | 68.4±1.8 |
| WarriorMath-DS-7B | DeepSeek-R1-Distill-Qwen-7B | 60.0±9.1 | 50.7±9.1 | 93.2±4.9 | 95.0±1.0 | 43.20±1.9 | 69.6±1.8 |

**Datasets.** To evaluate **WarriorMath**'s mathematical capabilities, we use six prevalent benchmarks: (1) **AIME 2024, AIME 2025** (AIME-2024, 2024; AIME-2025, 2025) are benchmarks that include particularly challenging math problems from the American Invitational Mathematics Examination (AIME) of 2024 and 2025, designed to assess advanced problem-solving skills. (2) **AMC 2023:** (AMC-2023, 2023) High school-level problems from the American Mathematics Competitions (AMC), testing core mathematical concepts (3) **MATH-500** (Lightman et al., 2024) is a dataset containing high school-level math problems. It serves to assess a model's ability to handle more advanced mathematical reasoning; and (4) **Minerva** (Lewkowycz et al., 2022) is a benchmark dataset comprising a diverse collection of quantitative reasoning problems that cover topics such as arithmetic, algebra, geometry, calculus, physics, and chemistry, with difficulty levels ranging from grade school to college. (5) **Olympiad Bench:** He et al. (2024a) an Olympiad-level bilingual scientific benchmark, featuring 8,476 problems from Olympiad-level mathematics and physics competitions. To ensure accurate evaluation, we follow the evaluation method proposed by Hochlehnert et al. (2025). We use the Pass@1 metric as the primary evaluation criterion. For each result, we report the mean and standard deviation computed over multiple random seeds. All experiments are conducted using lighteval (Fourrier et al., 2023) with the vllm backend (Kwon et al., 2023).

**Baseline.** We compare our method against reinforcement learning (RL) approaches trained on the Qwen2.5 Math Base models, including Oat-Zero (Liu et al., 2025a), LIMR (Li et al., 2025), and SimpleRL-Zoo (Zeng et al., 2025). We also evaluate our approach against supervised fine-tuning (SFT) baselines, such as s1.1 (Muennighoff et al., 2025), Eurus2 Prime (Cui et al., 2025), Bespoke Stratos (Bespoke Labs, 2024), OpenR1 (Face, 2025) and OpenThinker (Team, 2025). In addition, we consider recent state-of-the-art methods based on deepseek-r1-distill-qwen-7b as the backbone, such as LightR1 (Wen et al., 2025),and AReal-boba-RL-7b (inclusionAI, 2025).

**Implantation Details** During the data synthesis, we adopt 9 different generation configs where temperature $t \in \{0.60, 0.65, 0.70\}$ and top-p $p \in \{0.85, 0.90, 0.95\}$. The detailed prompts can be found in Appendix B.1. As for the training stage, the global batch size is set to 512, and the number of total training steps is set to 448. We use a learning rate of $1 \times 10^{-5}$ and a weight decay of $3 \times 10^{-7}$. Additionally, a WarmupLR scheduler with a warmup ratio of 0.2 is used.

## 4.2 PERFORMANCE AND COMPARISON

**Main Result.** The results on the math benchmarks are summarized in Table 1. WarriorMath achieves SOTA performance, with a pass@1 accuracy of 60% in AIME'24 and 56.7% in AIME'25, surpassing all other fine-tuned models. This highlights the efficacy of our approach in generating high-quality data and effective training process.

Table 2: Performance of different data synthesis strategies on three mathematical benchmarks.

| Models | Base | GSM8K | MATH-500 | AIME2024 |
|---|---|---|---|---|
| Qwen2.5-Math-7B-Instruct Yang et al. (2024) | - | 95.2 | 83.6 | 13.3 |
| Openmathinstruct-7B Toshniwal et al. (2025) | Qwen2.5-Math-7B | 92.0 | 79.6 | 10.0 |
| NuminaMath-7B Li et al. (2024c) | Qwen2.5-Math-7B | 92.9 | 81.8 | 20.0 |
| Evol-Instruct-7B Luo et al. (2025a) | Qwen2.5-Math-7B | 88.5 | 77.4 | 16.7 |
| KPDDS-7B Huang et al. (2025) | Qwen2.5-Math-7B | 89.9 | 76.0 | 10.0 |
| PROMPTCOT-Qwen-7B Zhao et al. (2025) | Qwen2.5-Math-7B | 93.3 | **84.0** | 26.7 |
| WarriorMath-Qwen-7b-SFT | Qwen2.5-7B | **95.7** | 83.8 | **36.7** |

Table 3: The proportion of different tasks in the training data.

| Mathematics Domain | Percentage (%) | Definition |
|---|---|---|
| Applied Mathematics | 11.4 | Apply mathematical methods to solve cross-field problems. |
| Algebra | 30.3 | Study of mathematical symbols and manipulation rules. |
| Discrete Mathematics | 12.9 | Study of discrete mathematical structures (e.g., graphs, integers). |
| Geometry | 14.7 | Study of properties/relations of points, lines, surfaces, solids. |
| Number Theory | 13.1 | Study of integers and integer-valued functions. |
| Precalculus | 1.2 | Math preparation for calculus (functions, trigonometry). |
| Calculus | 1.8 | Study of continuous change (derivatives, integrals). |
| Differential Equations | 0.5 | Equations involving derivatives for quantity change. |

Table 4: Results with Iterative Alignment.

| Model | AIME'24 |
|---|---|
| WarriorMath-Qwen-7b-SFT | 36.7±4.5 |
| *Iteration 1* | 42.5±5.5 |
| *Iteration 2* | 44.7±7.2 |
| *Iteration 3* | **48.3±2.6** |

Table 5: Results when learning from varying numbers of committee members.

| #Num | AIME'24 | AIME'25 | AMC'23 |
|---|---|---|---|
| 1 | 19.7±2.9 | 15.7±2.7 | 59.5±4.5 |
| 2 | 29.4±3.2 | 23.5±4.2 | 69.3±3.6 |
| 5 | 48.3±2.6 | 36.5±5.0 | 83.0±2.5 |

**Data Quality.** In order to assess the effectiveness of the problem generation pipeline, we compare our method with the following problem generation baselines: (1) **Evol-Instruct**: This method (Luo et al., 2025a) aims to enhance the quality of instruction data by improving both its complexity and diversity, thus facilitating the generation of more varied and challenging problems; (2) **KPDDS**: A data synthesis framework (Huang et al., 2025) that generates question-answer pairs by leveraging key concepts and exemplar practices derived from authentic data sources; (3) **OpenMathInstruct**: This method (Toshniwal et al., 2025) utilizes few-shot learning to prompt an LLM to create new math problems based on existing examples, without explicit instructions for adjusting difficulty or introducing new constraints; (4) **NuminaMath**: This approach (Li et al., 2024c) uses an LLM to generate novel math questions starting from a reference problem; (5) **PROMPTCOT**: This method (Zhao et al., 2025) synthesizes complex problems based on mathematical concepts and the rationale behind problem construction, emulating the thought processes of experienced problem designers. For fair comparisions we follow the evaluation scripts provided in (Zhao et al., 2025). The results of data quality assessment, presented in Tables 2, Our method achieves state-of-the-art performance across multiple benchmarks, outperforming the baselines, which highlights the efficacy of our defect-aware approach in generating high-quality problems

**Iterative Defect Alignment.** As a seed model $M_0$ we use the WarriorMath-Qwen-7b-SFT, which is fine-tuned with instruction data generated by Defect-Aware Committee Assessment. In each iteration, we generate $N = 32$ solutions per problem using sampling with temperature 0.7 and top-p 0.8, and verify the answer to select wrong solution $(c_i, y_i)$ in the loser set $G_i^l$. Then we generate $K = 10$ pairs per problem for training with our loss in Equation 3.2. In total, we perform three iterations, producing models $M_1, M_2, M_3$. The coefficient $\alpha$ is tuned in $\{0.25, 0.5, 1\}$ when training $M_1$, and we end up using 1 for all experiments in the paper. The coefficient $\beta$ in the DPO loss is tuned in $\{0.05, 0.1, 0.5\}$, and we end up using 0.1 in this experiment.

Overall results are given in Table 4. We find that WarriorMath outperforms supervised fine-tuning (SFT) on the gold (dataset-provided) data, and steady growth over the iteration rounds.

Table 6: Ablation results across six mathematical reasoning benchmarks. We report Pass@1 accuracy (mean $\pm$ std) across six reasoning benchmarks, consistent with the setup in Table 1.

| Models | AIME'24 | AIME'25 | AMC'23 | MATH500 | Minerva | Olympiad |
|---|---|---|---|---|---|---|
| WarriorMath-Qwen-7B-SFT | 36.7±4.5 | 36.5±5.0 | 83.0±2.5 | 88.3±1.4 | 41.2±2.8 | 52.1±0.8 |
| WarriorMath-Qwen-7B-SFT (w/o defect) | 32.0±5.9 | 21.4±4.3 | 74.1±5.2 | 86.6±0.6 | 31.7±0.6 | 51.9±0.4 |
| WarriorMath-Qwen-7B-SFT (w/o elo rating) | 35.4±5.6 | 24.0±4.1 | 79.5±5.1 | 86.7±1.6 | 38.4±2.2 | 50.1±2.3 |

## 4.3 ABLATION STUDY.

### 4.3.1 NUMBER OF COMMITTEE MEMBERS

Table 5 presents the results observed when the target model learns from varying numbers of committe members. The target model shows a significant improvement when learning from just one mathematical LLM, indicating that even a single committe member enables it to acquire a specific set of knowledge. However, as the number of members increases, **WarriorMath** benefits from learning across all mathematical LLMs. As a result, the model trained with 5 math LLMs outperforms others across all 6 benchmarks, demonstrating the advantages of integrating knowledge from multiple specialized experts.

### 4.3.2 DEFECT FILTERING

Our *defect-aware filtering* specifically targets problems proposed by teacher models, ensuring that the resulting data remain sufficiently challenging for downstream learners (e.g., Qwen2.5-Math, Qwen-DeepSeek-R1-Distill-7B). As base models grow more capable, many mined problems may be trivial; thus, defect-aware filtering becomes indispensable. Recent works (Muennighoff et al., 2025; He et al., 2025a; Zhang et al., 2025) independently adopt performance-based difficulty screening, further validating its effectiveness. To empirically assess the role of *base model*, we ablated the Defect-Aware Committee Assessment stage by removing them. Results in Table 6 clearly demonstrate that Defect Filtering with base model are crucial for optimal performance across benchmarks.

### 4.3.3 ELO RATING

Table 6 demonstrates the impact of the Judge system. Replacing the judge's Elo rating with mediate voting drops AIME'25 performance from 36.5% to 21.4%. Mediate voting alone suffer from high variance (noise from single judges), whereas Elo provides global consistency, representing more robust and accurate measure of a model's overall ability.

### 4.3.4 DATA ANALYSIS

**Data Dependence**  As discussed in Section 3.1.1, our data sources are derived from multi-expert LLMs, which distinguishes them from the commonly used datasets in existing research. To assess the novelty and dependence of our data, we conducted an analysis across all datasets. Figure 3 illustrates the overlap between the instructions mined from expert LLMs and those in two widely adopted math training datasets: (1) DeepScaleR (Luo et al., 2025b) and (2) Omni-MATH (Gao et al., 2025), measured by the ROUGE score. The majority of the mined instructions show a ROUGE score below 0.3, indicating significant distinctiveness from the existing datasets. Notably, no mined instruction exceeds a ROUGE score of 0.6, further confirming that these instructions are generated from the internal distribution of expert LLMs, rather than being simple replications or extensions of the training data. This unique source ensures a higher degree of independence, making the instructions especially valuable for model training by providing novel examples that can enhance the model's capabilities.

**Data Diversity**  A key characteristic of our training data is its extensive topical diversity across mathematical domains. As shown in Table 3, the classification results highlight that WarriorMath covers a wide range of core mathematical fields. These include foundational subjects like Applied Mathematics and Basic Geometry, as well as more advanced areas such as Number Theory and Differential Equations. This broad topical range ensures that models trained on WarriorMath are exposed to a rich variety of mathematical concepts and problem-solving strategies, enabling the development of more robust and versatile mathematical capabilities.

## 5 CONCLUSION

In this work, we present WarriorMath, a defect-aware framework that enhances mathematical ability in LLMs through defect-aware data synthesis and progressive training. Our method constructs high-quality, defect-aware training data by leveraging a committee of expert LLMs to generate, critique, and refine problems specifically designed to expose the base LLM's inherent defects. Through a two-stage progressive training process, WarriorMath incrementally aligns the model to stronger mathematical ability. Extensive experiments on six mathematical benchmarks demonstrate that WarriorMath achieves state-of-the-art performance among open-source models, highlighting the importance of learning from model-specific defects.

## 6 LIMITATION

This paper allows for the low-cost generation of high-quality and diverse data from scratch. However, as the number of expert models increases, the evaluation process becomes increasingly time-consuming. Designing more efficient and scalable multi-agent collaboration mechanisms remains an important direction for future research. Another limitation concerns the coarseness of the defect signal. Our current definition of a defect is a pragmatic, binary signal based solely on the final answer's correctness. This prevents the framework from providing the most fine-grained feedback.

## ETHICS STATEMENT

This work does not involve human subjects, personally identifiable information, or sensitive data. The datasets used are publicly available, and all experiments comply with the ICLR Code of Ethics.

## REPRODUCIBILITY STATEMENT

We have made extensive efforts to ensure reproducibility. The detailed methodology of our proposed approach is presented in Section 3, while the experimental settings, including training procedures and evaluation protocols, are described in Section 4. To further support reproducibility, we plan to release the complete source code and instructions upon the acceptance of this paper.

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

Table 7: **Evaluation results of 7B committee models.**

| Models | Base | AIME'24 | AIME'25 | AMC'23 | MATH500 | Minerva | Olympiad |
|---|---|---|---|---|---|---|---|
| *Qwen-Based Models* | | | | | | | |
| Qwen2.5-Math-7B-Base Yang et al. (2024) | - | 20.7±3.8 | 8.7±3.9 | 56.2±5.7 | 64.3±0.5 | 17.3±1.9 | 29.0±0.5 |
| Qwen2.5-Math-7B-Instruct (Yang et al., 2024) | Qwen2.5-Math-7B | 15.7±3.9 | 10.7±3.8 | 67.0±3.9 | **82.9±0.1** | 35.0±0.6 | 41.3±0.9 |
| Qwen2.5-7B-Instruct (Qwen et al., 2025) | Qwen2.5-7B | 12.3±3.2 | 7.3±3.4 | 52.8±4.8 | 77.1±1.2 | 34.9±1.0 | 38.7±1.0 |
| s1.1-7B (Muennighoff et al., 2025) | Qwen2.5-7B | 19.0±3.2 | 21.0±5.5 | 59.5±3.7 | 80.8±0.6 | 37.5±1.1 | **48.2±1.4** |
| Eurus-2-7B-PRIME (Cui et al., 2025) | Qwen2.5-7B | 17.8±2.2 | 14.0±1.7 | 63.0±3.9 | 80.1±0.1 | 37.5±1.0 | 43.9±0.3 |
| WarriorMath-Qwen-7B-Small | Qwen2.5-7B | **24.4±3.3** | **22.3±5.2** | **70.6±3.9** | 81.6±0.5 | **37.9±3.8** | 46.7±1.3 |

## A  DISCUSSION

### A.1  DIFFERENCES FROM OTHER DATA SYNTHESIS STRATEGIES

WarriorMath diverges from conventional knowledge distillation in three fundamental ways:

**(1) Data Sources.** Traditional distillation typically relies on pre-existing seed datasets (e.g., NuminaMath-1.5 (Li et al., 2024c), DeepScaleR (Luo et al., 2025b)) and then distills solutions from expert models. In contrast, WarriorMath elicits new, challenging problems directly from experts and derives high-quality solutions through competitive interactions among them.

**(2) Learning Objectives.** While conventional distillation primarily focuses on imitating expert performance, WarriorMath establishes a closed-loop paradigm of *detecting defects and correcting weaknesses*, where the base model's shortcomings are explicitly identified and iteratively addressed.

**(3) Knowledge Integration.** Standard distillation usually adopts responses from a single expert, whereas WarriorMath aggregates strengths from multiple experts (e.g., Phi, QwQ) and adaptively balances their influence via an Elo rating mechanism.

### A.2  EFFECTIVENESS WITH SIMILAR-SIZED MODELS

To verify that WarriorMath does not rely solely on distillation from larger models, we conducted the experiment using a committee of similar-sized models (7B). We select three models:Qwen2.5-7B-Instruct, Qwen2.5-Math-7B-Instruct, DeepSeek-R1-Distill-Qwen-7B to train the Qwen2.5-7B-Base.

As shown in Table 7, The model trained by the 7B-Committee achieved 24.4% on AIME24, which outperforms the teacher model Qwen2.5-Math-7B-Instruct. This demonstrates that WarriorMath is effective even without larger "teacher" models. Notably, baselines in Table 7 (e.g., s1.1) rely on distillation from stronger teachers (including Gemini and DeepSeek-R1-671B, which outperform our strongest teacher model). Despite this advantage, these baselines still underperform WarriorMath. We attribute this improvement to two key factors: **1.Diversity:** Different models have different knowledge boundaries. An "ensemble" of 7B models covers more ground than a single one. **2.Verification >** **Generation:**A 7B model can often verify the correctness of a solution (acting as a Judge) or generate a hard problem (acting as an Examiner) that is difficult for itself or peers to solve directly. This allows the system to bootstrap improvement rather than just distilling knowledge.

### A.3  COMPUTATIONAL COST AND EFFICIENCY

We now report the full computational profile: **Committee assessment:** 8×A100 GPUs for 96 hours, producing 240M tokens (200K samples). **Training:** 8×A100 GPUs for 160 GPU hours (batch size 512, 448 steps). This is ~30% more efficient than baselines such as LIMR, which required over 3,000 hours.

### A.4  CLARIFYING BASELINE FAIRNESS

In Table 1, baselines such as s1.1 (Muennighoff et al., 2025), Light-R1 (Wen et al., 2025), and LIMR (Li et al., 2025) rely on distillation from stronger teachers, including Gemini and DeepSeek-R1 671B (DeepSeek-AI et al., 2025), which surpass our strongest teacher (DeepSeek-R1-Distill-Llama-70B). Despite this advantage, these baselines underperform WarriorMath. For instance, on AIME-2024, WarriorMath-Qwen-7B achieves 48.3% Pass@1, compared to LIMR's 30.7% and s1.1's

19.0%. This gap underscores the superiority of WarriorMath's synthesis paradigm over conventional distillation.

In Table 2, we ensure fair comparisons across data generation methods. For approaches without released datasets (e.g., Evol-Instruct (Luo et al., 2025a), KPDDS (Huang et al., 2025)), we replicate their setups using Llama-3.1-70B-Instruct (Llama Team, 2024) to generate problems at scale matched to WarriorMath. For NuminaMath (Li et al., 2024c) and OpenMathInstruct (Toshniwal et al., 2025), we directly adopt their released sets. To standardize solution generation, we uniformly employ Qwen2.5-Math-72B-Instruct (Qwen et al., 2025) as the solver across all baselines.

### A.5   ON THE RISK OF DATA LEAKAGE

All WarriorMath problems are synthesized by expert models through adversarial prompting, rather than sampled from existing datasets. If leakage were to occur, it would imply that the expert models themselves had memorized training data. However, the experts employed (e.g., DeepSeek-R1, Qwen2.5-Math) explicitly document rigorous data curation efforts to mitigate leakage risks (DeepSeek-AI et al., 2025; Qwen et al., 2025).

To further ensure novelty, we apply ROUGE-based semantic similarity screening (Figure 3) to reduce overlap. Consequently, WarriorMath instructions provide diverse, independent training examples that enhance generalization.

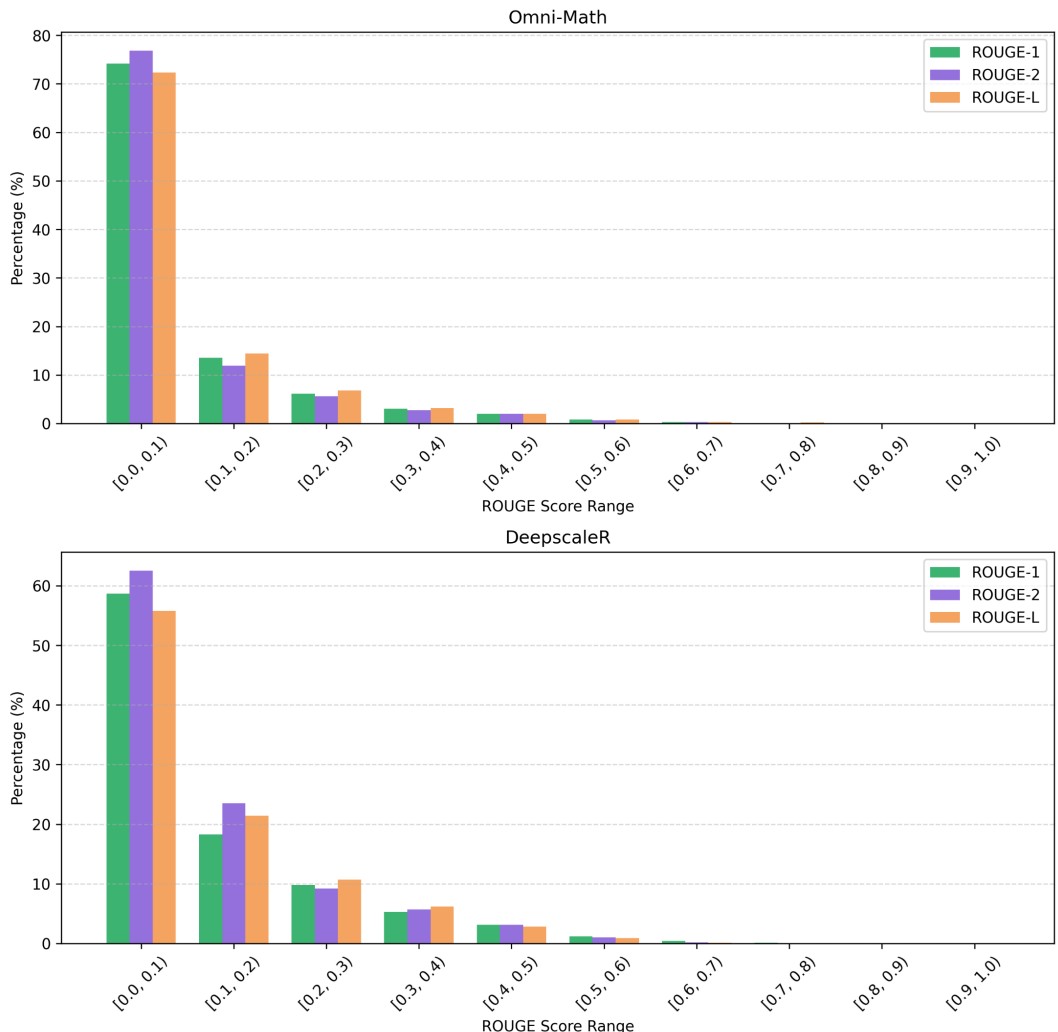

Figure 3: The overlapping rate between the mined instructions and existing training datasets.

# B INSTRUCTION DATA SYNTHESIS

## B.1 PROMPTS FOR INSTRUCTION MINING

---

**Instruction Mining Prompt**

**Prompt:**
Please act as a world-class expert in designing extremely challenging and diverse math problems. Your goal is to create problems that thoroughly test a model's reasoning abilities by inducing a variety of potential failure modes (e.g., reasoning, understanding, calculation, or strategy errors).
For each problem you design, please ensure the following:

1. **Quality:** Questions must be well-formatted, clearly structured, and unambiguous.

2. **Difficulty:** Problems should require deep mathematical reasoning and not be solvable via simple pattern recognition or surface-level heuristics.

3. **Diversity:** Problems must span a wide range of mathematical domains, such as algebra, calculus, discrete math, geometry, and others.

4. **Challenge:** Each problem should be adversarially constructed to trigger potential weaknesses in advanced models, such as subtle traps or misleading intermediate steps.

Always provide the final answer enclosed within \boxed{} for clarity.

---

Figure 4: The prompt for generating challenging and diverse math problems.

## B.2 CASE STUDY OF MINING PROBLEMS

---

**Examples of mining problems**

**Case #1**: Let $S$ be the set of all convex quadrilaterals inscribed in the circle $x^2 + y^2 = 25$ with vertices at points having integer coordinates. Determine the maximum possible area of such a quadrilateral $Q$, given that the sum of the $x$-coordinates of its vertices equals the sum of the $y$-coordinates. Express your answer as a reduced fraction $\frac{m}{n}$, where $m$ and $n$ are coprime positive integers, and find $m + n$.

---

**Case #2**: Consider triangle $ABC$ with $AB = 13$, $BC = 14$, and $AC = 15$. Let $O$ be the circumcenter and $H$ the orthocenter. Let the circumradius be $R$. A circle centered at $O$ with radius $R/2$ intersects the nine-point circle at points $P$ and $Q$. Find the length of $PQ$.

---

**Case #3**: Consider the function $f(x, y) = x^3 + y^3 - 3xy$. Find the maximum value of $f(x, y)$ on the closed disk $x^2 + y^2 \leq 1$.

---

**Case #4**: Consider a sequence of $n$ independent coin flips, where each flip has a probability $p$ of landing heads and a probability $q = 1 - p$ of landing tails. Let $X$ be the random variable representing the number of heads in the sequence. Find the probability that $X$ is even.

---

**Case #5**: Let $f(x) = x^3 - 3x + 1$. The polynomial $f(x)$ has three real roots, denoted by $\alpha, \beta, \gamma$. Define the sequence $\{a_n\}$ by $a_1 = \alpha + \beta + \gamma$, $a_2 = \alpha^2 + \beta^2 + \gamma^2$, and for $n \geq 3$, $a_n = \alpha^n + \beta^n + \gamma^n$. Find the value of $a_{2023}$ modulo 3.

---

**Case #6**: Consider the set $S = \{1, 2, 3, \ldots, 100\}$. A subset $A$ of $S$ is called "sum-free" if there do not exist distinct elements $a, b, c \in A$ such that $a + b = c$. Determine the maximum possible number of elements in a sum-free subset of $S$.

---

**Case #7**: A fair six-sided die is rolled repeatedly until a 6 appears. Let $X$ be the number of rolls required. Define the function $f(n)$ as the probability that $X$ is a multiple of $n$. Find the value of $f(3)$.

---

**Case #8**: Determine the value of $a_{100}$ for the sequence defined by $a_1 = 1$ and $a_{n+1} = a_n + \gcd(a_n, n)$ for $n \geq 1$.

---

Figure 5: Examples of mining problems.

B.3   PROMPT FOR PROBLEM QUALITY EVALUTION AND DISCARDED PROBLEMS EXAMPLES

---

**Math Instruction Quality Evaluation Prompt**

You are a senior university-level mathematics instructor with extensive expertise in advanced topics such as Algebra, Precalculus, Number Theory, Geometry, and Combinatorics. Your task is to **evaluate the quality of mathematical problem statements** based on their clarity, formatting, conceptual soundness, computational complexity, and contextual relevance. Each problem should be scored on a scale of **1 to 10**, and your output must follow a structured JSON format. **Evaluation Criteria:**

- **Clarity and Completeness:** Is the problem clearly stated without ambiguity? Are all necessary variables, conditions, and constraints defined? Is the mathematical notation properly used and well-structured?

- **Conceptual Soundness and Difficulty:** Does the problem involve meaningful, non-trivial mathematical reasoning or advanced concepts? Does it promote critical thinking and apply appropriate mathematical principles?

- **Computational Complexity:** Does the solution process require more than basic arithmetic or trivial computation? Are there non-obvious calculations, transformations, or logical deductions involved?

- **Contextual Relevance and Verifiability:** Is the problem well-grounded in a practical, educational, or theoretical context? Is the problem solvable or verifiable using existing tools and methods? Avoid problems that are vague, proof-based without criteria, or ill-posed.

**Scoring Scale:**

- **Excellent (9–10):** The instruction is verifiable, properly formatted, and conceptually sound.

- **Good (6–8):** Minor issues in clarity or formatting. Still verifiable and mathematically valid.

- **Average (3–5):** Noticeable flaws in clarity, completeness, or relevance.

- **Poor (1–2):** Ambiguous, improperly defined, unverifiable, or conceptually flawed.

**Your Output Format:** Your output must be a **JSON list**, where each element is a dictionary with the following keys:

- `instruction`: The original math problem.

- `score`: An integer from 1 to 10 representing your evaluation.

- `reason`: A detailed explanation justifying the score based on the criteria above.

---

Figure 6: Prompt used to evaluate the quality of mathematical instructions, including scoring criteria and output format. Only instructions rated 6 or higher are considered suitable for use in further steps.

---

**Examples of discarded problems**

We provide examples of low-quality problems that were filtered out during problem selection, categorized according to our criteria (Verifiability, Proper Formatting, Clarity):

- **(1) Unverifiable Problem:**

  ```
  Write a python code that run a math function like "log(base ,
  number)".
  ```

  *Reason: This problem is not a mathematical question with a concrete answer, and cannot be automatically evaluated.*

- **(2) Poor Formatting:**

  ```
  With what polynomial function equation do you want to calculate the
  vertex of the graph?
  f(x)=2x2-x3+5x4
  ```

  *Reason: The input mixes natural language with improperly rendered HTML, leading to parsing and readability issues.*

- **(3) Incomplete Problem:**

  ```
  Chef's portion took 15 seconds, and the assistant's portion took 45
  seconds.
  ```

  *Reason: The question is incomplete and lacks a clear task or objective for the model to solve.*

Figure 7: Examples of discarded problems during the filtering process.

## B.4 VOTE AND ELO RATING

The reviewer B is required to respond to the Examiner's question, while the Examiner A must also provide an answer to its own instruction. Then we can calculate the *local score* for each response:

$$x^i_{A>B} = \frac{t_A}{t_A + t_B} \quad x^i_{B>A} = \frac{t_B}{t_A + t_B} \tag{8}$$

where $x^i_{A>B}$ and $x^i_{B>A}$ are the local scores for A's and B's responses to the instruction $i$. $x^i_{A>B}$ represents the percentage of votes that candidate A receives, while $x^i_{B>A}$ similarly represents the percentage of votes that candidate B receives. $t_A$ and $t_B$ are the number of votes which A and B win.

However, relying solely on the *local score* to select the winner can be problematic. In some cases, a weaker model may receive more votes than a stronger one, even though its responses are not significantly better. This can occur because the *local score* may not fully capture the quality of the model's performance, especially in situations where the voting is influenced by factors, such as randomness or bias from LLM judges.

To address this limitation, we propose considering both local contingency and global consistency in the decision-making process. Instead of directly basing our analysis on the immediate voting outcomes, we introduce the concept of the *global score* — specifically, the Elo rating , which provides a more comprehensive reflection of a model's relative performance over time and across various evaluations. The Elo rating system, originally developed to calculate the relative skill levels of players in two-player games (such as chess), has been successfully adapted to assess the performance of competitors in a range of competitive scenarios, including esports and other skill-based games.

By incorporating the Elo rating, we account for both local performance in individual contests and global performance across multiple rounds, providing a more robust and accurate measure of a model's overall ability. This helps to mitigate the risk of weak models winning based on isolated, potentially unrepresentative votes:

$$X^{Elo}_{A>B} = \frac{1}{1 + 10^{(R_B - R_A)/400}}$$
$$X^{Elo}_{B>A} = \frac{1}{1 + 10^{(R_A - R_B)/400}} \tag{9}$$

where $X^{Elo}_{A>B}$ and $X^{Elo}_{B>A}$ indicate the expected probabilities of A defeating B and B defeating A, respectively. $R_A$ and $R_B$ are the Elo rating of A and B, which are updated dynamically and iteratively. Given the battle result of A and B on an instruction $i$, we update them by:

$$R_A \leftarrow R_A + K \times (s^i_{A>B} - X^{Elo}_{A>B})$$
$$R_B \leftarrow R_B + K \times (s^i_{B>A} - X^{Elo}_{B>A}) \tag{10}$$

where $s^i_{A>B}$ and $s^i_{B>A}$ are the actual score of the battle result of player A and B (1 for a win, 0.5 for a draw, and 0 for a loss). The factor $K$ controls the sensitivity of rating changes.

Based on Equation 8 and Equation 9, we can obtain the final score of A's response for instruction $i$:

$$e^i_A = \sum_{B \in Com \backslash A} \alpha X^{Elo}_{A>B} + (1 - \alpha) x^i_{A>B} \tag{11}$$

where $Com$ is the set of all the competitors and '\' is the subtraction operation. $\alpha$ is the coefficient to balance the local contingency and global consistency.

**Pair-wise Answer Quality Evaluation Prompt**

**Prompt:**
Please act as an impartial judge and evaluate the quality of the response provided by an AI assistant to the user prompt displayed below.
You will be given a user prompt, a reference answer, and the assistant's answer. Your job is to compare the assistant's answer with the reference one and assign a score.
For each user prompt, carry out the following steps:

1. Consider if the assistant's answer is helpful, relevant, and concise.
   - **Helpful** means the answer correctly responds to the prompt or follows the instructions.
   - **Relevant** means all parts of the response closely connect or are appropriate to what is being asked.
   - **Concise** means the response is clear and not verbose or excessive.
2. Then consider the creativity and novelty of the assistant's answer when needed.
3. Identify any missing important information in the assistant's answer that would be beneficial to include when responding to the user prompt.
4. After providing your explanation, you must rate the assistant's answer on a scale of 1 to 10, where a higher score reflects higher quality.

**Guidelines for Scoring:**
- **Assistant's Answer >> Reference Answer (7–10):** The assistant's answer is significantly or slightly better than the reference answer.
- **Assistant's Answer == Reference Answer (5–6):** The quality of assistant's answer is relatively the same as that of the reference answer.
- **Assistant's Answer << Reference Answer (1–4):** The assistant's answer is significantly or slightly worse than the reference answer.

**User Prompt:**
{instruction}

**Reference Answer:**
{reference}

**Assistant's Answer:**
{response}

Use double square brackets to format your scores, like so: [[7]].

Figure 8: The prompt for the evaluation of pairwise comparison.

## C    THE USE OF LARGE LANGUAGE MODELS

We used a Large Language Model (LLM) only as a writing assistant to polish the language of the manuscript (*e.g.*, grammar refinement, style adjustment, and clarity improvement). The research ideas, methodology design, experiments, and analysis were entirely conceived, implemented, and validated by the authors without reliance on the LLM. The LLM did not contribute to research ideation, experimental design, or result interpretation.

