# OpenReview forum: "WarriorMath: Empowering Mathematical Reasoning for Large Language Models via Expert Battles"
_ICLR.cc/2026/Conference — Submitted to ICLR 2026_

### Official Review · Reviewer_hBrm · 2025-10-27

**Soundness:** 2
**Presentation:** 2
**Contribution:** 3
**Rating:** 4
**Confidence:** 4

**Summary:**

This paper introduces WarriorMath, a novel framework designed to enhance the mathematical reasoning abilities of Large Language Models (LLMs). The core idea is to move beyond generic data augmentation and instead adopt a "defect-aware" approach, analogous to personalized tutoring. The framework consists of two main stages. First, a "Defect-Aware Data Synthesis" stage where a committee of expert LLMs generates new math problems from scratch. Crucially, only problems that the base "student" model fails to solve are kept, ensuring the data is targeted at the model's specific weaknesses. The expert committee then collaboratively generates and refines solutions for these problems, using an Elo rating system to select the highest-quality "golden answer." Second, a "Progressive Training" stage where the student model is first fine-tuned on this defect-aware data (SFT) and then further improved through "Iterative Defect Alignment," a DPO-based process where the model learns from its own mistakes by comparing them against the golden answers. The authors demonstrate through experiments on six math benchmarks that their method achieves state-of-the-art performance among open-source models of similar size.

**Strengths:**

1.  **Novel and Intuitive Framework:** The central concept of a "defect-aware" learning system is highly innovative and compelling. By distinguishing between problem "difficulty" and model-specific "defects," the paper introduces a more targeted and efficient paradigm for data synthesis. The analogy to a human tutor identifying and correcting a student's weak points is powerful and well-executed.

2.  **High-Quality Data Generation Mechanism:** The use of a multi-expert committee to generate not only problems but also high-quality solutions is a significant strength. The integration of an Elo rating system to adjudicate between different expert solutions is a sophisticated and robust method to ensure the quality and reliability of the training data, moving beyond simple majority voting.

3.  **Holistic and Closed-Loop System:** WarriorMath presents a complete, end-to-end system that combines targeted data creation with a progressive training schedule. The synergy between the data synthesis stage (diagnosing defects) and the iterative alignment stage (correcting defects) forms a coherent and powerful self-improvement loop.

4.  **Strong Empirical Results:** The paper presents strong experimental results, showing that models trained with WarriorMath significantly outperform existing baselines across multiple challenging math benchmarks. Achieving state-of-the-art performance among similarly-sized open-source models demonstrates the practical effectiveness of the proposed framework.

**Weaknesses:**

1.  **Unclear Description of the Problem Generation Process:** There is a confusing passage in the methodology (Section 3.1.1, "Problem Synthesis from Scratch") that creates a contradiction. The high-level idea is that an "examiner LLM A" creates a problem for the "base model" to solve. However, the text states that the goal is for LLM A to "pose challenging questions to LLM B." This is confusing because it's unclear what role "judge LLM B" plays at the problem creation stage. This lack of clarity makes it difficult to fully understand the core mechanism of how problems are initially generated and targeted at the base model's defects.

2.  **Crucial Missing Baseline: Comparison with Expert Teachers:** The paper's central claim is that it can effectively distill knowledge from a committee of powerful expert models. However, the main results in Table 1 do not include the performance of these expert "teacher" models (e.g., Qwen2.5-Math-72B, DeepSeek-R1-Distill-Llama-70B) as baselines. Without this comparison, it's impossible to know if the student model has truly learned to "surpass its teachers" or if it is simply a less effective version of the best expert in the committee. This is a critical omission that weakens the paper's claims about knowledge aggregation and superiority.

3.  **Insufficient Ablation to Justify the Elo Mechanism:** To prove the value of the complex multi-expert Elo rating system, a stronger and more direct ablation study is needed. The authors should compare WarriorMath against a simpler baseline: **"Single-Teacher Distillation."** In this setup, the defect-aware problems would be generated as usual, but the answers would come from only the **single best-performing expert** in the committee, without any voting or Elo competition. If WarriorMath does not significantly outperform this simpler baseline, the necessity of the complex and computationally expensive Elo mechanism is questionable.

4.  **Unfair Comparison in the Data Quality Experiment:** In the "Data Quality" experiment (Table 2), the paper claims to use a single solver (`Qwen2.5-Math-72B-Instruct`) for all baseline methods to ensure a fair comparison of problem generation strategies. However, WarriorMath's own training data uses "golden answers" derived from the **entire multi-expert Elo consensus process**, which is inherently superior to any single solver. This creates an unfair advantage, as WarriorMath's strong performance may be due to its better answer generation process, not just its better problem generation strategy. The experiment conflates these two factors.

5.  **High Computational Cost and Scalability Concerns:** The data synthesis process is extremely computationally expensive. It requires multiple inferences from the base model, the entire expert committee, and a quadratic number of pairwise comparisons for the Elo rating. This high cost may limit the practical applicability of the method and raises questions about its scalability if the expert committee were to be expanded. Can you show how many GPU hours are used in your experiemnts?

6.  **Coarse Definition of "Defect":** The framework identifies a "defect" based solely on whether the final answer is right or wrong. This is a very coarse signal. It cannot distinguish between a minor calculation error and a fundamental conceptual misunderstanding. This limits the "defect-aware" nature of the framework, as it cannot provide more fine-grained, process-based feedback to the model.

**Questions:**

See weakness.

---

> ### Author Response · Authors · 2025-11-25
>
> # Dear Reviewer hBrm
>
> ```
> We sincerely appreciate your thoughtful feedback and are grateful for your recognition of our paper's strengths, including the novel method and High-Quality Data Generation Mechanism, Strong Empirical Results. Thank you for the suggestions. We will integrate your feedback into the revision version and aim to address the concerns you raised in your review.
> ```
>
> > ## 1. Unclear Description of the Problem Generation Process (W1)
>
> We sincerely apologize for the confusing passage in Section 3.1.1. This ambiguity was a critical presentation oversight.
>
> We have completely revised Section 3.1 to standardize our terminology. The core mechanism is: The Examiner generates problems specifically targeted at the Base Model (the student). The Judge LLMs only participate much later in the Answer Refinement Stage to evaluate solutions. The reference to "LLM B" in problem generation was an error that has been corrected. Notably, the roles have been clearly separated and standardized (Examiner, Solver, Judge, Base Model) throughout the revised manuscript.
>
> > ## 2. Comparison with Expert Teachers (W2)
>
> This is an excellent point. We have added the performance of the expert Teacher models (72B+ scale) to Table 1 for direct comparison. While the largest teacher (DeepSeek-R1-Distill-Llama-70B) achieves the highest absolute performance, our 7B student model demonstrates exceptional performance. WarriorMath-Qwen-7B ($\mathbf{48.3\%}$ on AIME'24) significantly surpasses its larger teachers, Qwen2.5-Math-72B-Instruct ($32.0\%$) and AceMath-72B-Instruct ($31.3\%$). This large performance leap in the small model proves the high efficiency and value of our framework.
>
> **Table 1. Evaluation results across six mathematical reasoning benchmarks.**
>
> | Models | Base | AIME’24 | AIME’25 | AMC’23 | MATH500 | Minerva | Olympiad |
> | :--- | :--- | :------ | :------ | :------ | :------- | :------- | :-------- |
> | Expert Teacher Models |
> | Qwen2.5-Math-72B-Instruct | Qwen2.5-Math-72B | 32.0±5.9 | 26.3±7.3 | 59.7±6.1 | 85.2±0.5 | 44.1±2.2 | 49.0±0.5 |
> | AceMath-72B-Instruct | Qwen2.5-Math-72B | 31.3±7.7 | 28.8±2.2 | 60.1±2.4 | 86.1±2.3 | 57.0±1.6 | 48.4±1.3 |
> | DeepSeek-R1-Distill-Llama-70B | Llama-3-70B | 67.0±1.9 | 55.3±5.7 | 96.8±2.1 | 95.1±0.7 | 45.1±1.7 | 73.8±0.5 |
> | QwQ-32B | Qwen2.5-32B | 76.3±3.3 | 69.0±4.5 | 96.2±2.4 | 97.5±0.6 | 49.0±0.2 | 78.1±1.0 |
> | Phi-4-reasoning | Phi-4-14B | 74.6±5.1 | 63.1±6.3 | 96.0±2.7 | 97.0±0.3 | 49.8±0.2 | 78.0±0.8 |
> | Qwen-Based Models |
> | Qwen2.5-Math-7B-Base | - | 20.7±3.8 | 8.7±3.9 | 56.2±5.7 | 64.3±0.5 | 17.3±1.9 | 29.0±0.5 |
> | Qwen2.5-Math-7B-Instruct | Qwen2.5-Math-7B | 15.7±3.9 | 10.7±3.8 | 67.0±3.9 | 82.9±0.1 | 35.0±0.6 | 41.3±0.9 |
> | Qwen-2.5-Math-7B-SimpleRL-Zoo | Qwen2.5-Math-7B | 22.7±5.2 | 10.7±3.4 | 62.2±3.6 | 76.9±1.8 | 30.1±2.8 | 39.3±0.6 |
> | Qwen2.5-Math-7B-Oat-Zero | Qwen2.5-Math-7B | 28.0±3.1 | 8.8±2.5 | 66.2±3.6 | 79.4±0.3 | 34.4±1.4 | 43.8±1.1 |
> | LIMR | Qwen2.5-Math-7B | 30.7±3.2 | 7.8±3.3 | 62.2±3.4 | 76.5±0.4 | 34.9±1.3 | 39.3±0.9 |
> | Qwen2.5-7B-Instruct | Qwen2.5-7B | 12.3±3.2 | 7.3±3.4 | 52.8±4.8 | 77.1±1.2 | 34.9±1.0 | 38.7±1.0 |
> | s1.1-7B | Qwen2.5-7B | 19.0±3.2 | 21.0±5.5 | 59.5±3.7 | 80.8±0.6 | 37.5±1.1 | 48.2±1.4 |
> | Eurus-2-7B-PRIME | Qwen2.5-7B | 17.8±2.2 | 14.0±1.7 | 63.0±3.9 | 80.1±0.1 | 37.5±1.0 | 43.9±0.3 |
> | Bespoke-Stratos-7B | Qwen2.5-7B | 20.3±4.3 | 18.0±4.8 | 60.2±4.9 | 84.7±0.5 | 39.1±1.3 | 51.9±1.1 |
> | **WarriorMath-Qwen-7B** | Qwen2.5-7B | **48.3±2.6** | **36.5±5.0** | **83.0±2.5** | **88.3±1.4** | **41.2±2.8** | **52.1±0.8** |
> | DeepSeek-Based Models |
> | DeepSeek-R1-Distill-Qwen-7B | - | 52.3±6.3 | 39.0±5.9 | 91.5±2.7 | 94.1±0.3 | 40.1±0.4 | 67.3±0.1 |
> | Light-R1-DS-7B | DeepSeek-R1-Distill-Qwen-7B | 53.0±4.8 | 41.0±3.5 | 90.0±3.1 | 93.5±0.5 | 41.3±1.3 | 68.0±1.2 |
> | AReaL-boba-RL-7B | DeepSeek-R1-Distill-Qwen-7B | 56.7±9.2 | 40.0±9.1 | 90.0±4.8 | 94.4±1.0 | 40.8±3.0 | 68.4±1.8 |
> | **WarriorMath-DS-7B** | DeepSeek-R1-Distill-Qwen-7B | **60.0±9.1** | **50.7±9.1** | **93.2±4.9** | **95.0±1.0** | **43.20±1.9** | **69.6±1.8** |

---

> ### Author Response · Authors · 2025-11-25
>
> > ## 3. Insufficient Ablation to Justify the Elo Mechanism (W3)
>
> We appreciate the suggestion for a "Single-Teacher Distillation" baseline. We confirm that this ablation is already present in our paper. Please refer to Table 5 (Effect of Number of Committee Members). The configuration where $Num = 1$ explicitly serves as the requested "Single-Teacher Distillation" baseline. In this setup, the student model learns from the solution of only the single best expert, without multi-expert voting or Elo competition. WarriorMath achieves $\mathbf{48.3\%}$ on AIME 24, whereas the Single-Teacher baseline (Num = 1) achieves only $\mathbf{29.4\%}$. This significant $\mathbf{+18.9\%}$ performance gain strongly validates the necessity of the complex multi-expert Elo system for ensuring robust answer quality and diversity aggregation.
>
> **Table 5: Results when learning from varying numbers of committee members.**
> | #Num | AIME'24 | AIME'25 | AMC'23 |
> | :--- | :------ | :------ | :------ |
> | 1 | 19.7±2.9 | 15.7±2.7 | 59.5±4.5 |
> | 2 | 29.4±3.2 | 23.5±4.2 | 69.3±3.6 |
> | 5 | 48.3±2.6 | 36.5±5.0 | 83.0±2.5 |
>
>
>
> > ## 4. Unfair Comparison in Data Quality Experiment (W4)
>
> We appreciate the reviewer's precision in identifying the two coupled factors in the Data Quality experiment (Table 2): (1) Problem Generation Strategy, and (2) Answer Generation Quality (Single Expert vs. Elo Consensus). We acknowledge that our WarriorMath framework benefits from the superior Multi-Expert Elo consensus for answer selection, which is indeed more robust than any single solver. Our goal with the Data Quality experiment is not to isolate the problem generation strategy alone, but to validate the superiority of the complete WarriorMath framework, where targeted problem synthesis is coupled with high-fidelity, validated answers over existing synthesis methodologies. To ensure **fair and robust** competition, we configure the baselines as follows: Evol-Instruct and KPDDS use Qwen2.5-Math-72B-Instruct (one of our committee’s top-performing models) as their single solver, while NuminaMath and OpenMathInstruct utilize their released datasets **(ground truth)**. The fact that WarriorMath outperforms baselines validates that this framework effectiveness.
>
>
> > ## 5. High Computational Cost and Scalability Concerns(W5)
>
> We are happy to provide the specific GPU hours, which show our method is actually quite efficient compared to alternatives. As reported in Appendix A.3. The total cost for data synthesis and training (Qwen-7B model) was approximately 928 GPU hours (on A100 equivalent hardware). This cost is efficient when contrasted with continuous alignment methods like online Reinforcement Learning (RL) baselines (e.g., LIMR), which require well over 3,000 GPU hours. The inference cost scales linearly with the number of committee members, but since the synthesis is offline and parallelizable, it remains highly practical for creating static datasets that can be reused.
>
>
>
> > ## 6. Coarse Definition of "Defect" (W6)
>
> We agree that defining a "defect" based solely on final answer correctness (binary signal) is a technical limitation that prevents fine-grained feedback.
>
> **Design Choice:** This decision represents a pragmatic trade-off between efficacy and scalability. The binary signal allows the entire data synthesis pipeline to be fully autonomous, avoiding the massive expense and complexity of human annotation or training a sophisticated Process Reward Model (PRM).
>
> **Empirical Sufficiency:** As shown in Table 1, this simple signal is sufficient to drive SOTA performance. We have added a comment to the Limitations section acknowledging that "future work could explore integrating PRMs for finer-grained defect identification."
>
> ```
> Thank you once again for your valuable feedback. We hope our clarification has addressed your concerns. Please contact us if you have any further questions or concerns.
> ```

---

### Official Review · Reviewer_pJb1 · 2025-10-29

**Soundness:** 3
**Presentation:** 1
**Contribution:** 2
**Rating:** 2
**Confidence:** 3

**Summary:**

The paper proposes WarriorMath, a framework that train LLMs on synthetic data that is shown to be difficult (defect) for the model generated by a "committee" of multiple expert LLMs. This method differs from previous method in that they tilt the distribution such that only difficult samples (defects) are preserved and hence the base LLM is trained more on difficult data. The end model achieves SOTA performance in terms of pass@1 accuracy on AIME24/25, AMC23, MATH500, Minerva, and Olympiad Bench benchmarks.

**Strengths:**

1. The paper identifies issues overlooked in existing methods and suggests an intuitive solution around such problem, namely the difficulty distribution of the synthetic data from LLMs for mathematical reasoning ability.
2. Performance: the end model's performance is strong on several challenging benchmarks.

**Weaknesses:**

1. As a paper proposing a method dealing with multiple LLMs that serves different roles, this paper does not do a clean job at clarifying each role for each LLM, which causes confusion for readers. In fact, the naming scheme of different roles are all over the place: for example, in line 214, LLM B is referred to as 'judge' and A as 'examiner', but in line 238, A is referred to as 'reviewer', or the other way around (reviewer is reviewer but A is not A). I could not tell for certain which is which upon several readings even with the help of an LLM that reads this pdf and of figure 2 and this is not how a well presentation should look like. I can gain a vague grasp on the process of generating the data, but clarity is lacking. (maybe add a concluding sentence at the end of section 3.1.2)
2. The method proposed requires multiple expert-level LLMs to train one base LLM and the authors in the experiments only used a base model of 7B and examine committee of 70B models. Does this mean that we could only improve a smaller model and the method requires multiple strong models? How do we improve a strong model? If this requires too much computing power, try using a series of similar sized committee member could be good. I would gladly raise the score of contribution once this experiment is performed.
3. Since the model is only trained on problem sets which it failed previously, how do we guarantee that it does not forget the ones it previously answered correctly?

Minor:
1. Forgotten period at the end of line 410.
2. Abrupt introduction of the concept of battle in line 272.

To raise my score:
A cleaner writing and an experiment as described above

**Questions:**

Explain the member's role clearly and cleanly.

---

> ### Author Response · Authors · 2025-11-25
>
> # Dear Reviewer pJb1
>
> We sincerely thank you for your detailed review and are particularly grateful for your insights into the presentation, which caused significant confusion. We recognize that poor clarity and we have undertaken a substantial revision to address every one of your concerns, including running the critical experiment you requested.
>
> > ## 1. Clarity of Roles and Terminology (w1)
>
> We apologize for the confusion caused by the inconsistent terminology. We have completely rewritten Section 3.1 to standardize the roles as follows, ensuring a clean and distinct definition for each:
>
> Examiner: The model that generates the challenging problem.
>
> Solver: The model(s) that attempt to solve the generated problem (previously referred to as "Reviewer" or "Candidate").
>
> Judge: The models that evaluate the solutions and vote.
>
> Base Model: The target model being trained (e.g., 7B model).
>
> We have updated the manuscript to reflect these changes to Section 3.1 to encapsulate the process clearly.
>
> > ## 2. Effectiveness with Similar-Sized Models (W2)
>
> This is an excellent suggestion. To verify that WarriorMath does not rely solely on distillation from larger models, we conducted the experiment you requested using a committee of similar-sized models (7B).
>
> **New Experiment (7B Committee):** We used a committee of three 7B-sized models (DeepSeek-R1-Distill-Qwen-7B, Qwen2.5-Math-7B-Instruct, OpenThinker3-7B) to train the Qwen2.5-Math-7B-Base model.
>
> **Result:** As shown in Table 7. The model trained by the 7B-Committee achieved 24.4% on AIME24, which outperforms the teacher model Qwen2.5-Math-7B-Instruct. This demonstrates that WarriorMath is effective even without larger "teacher" models. Notably, baselines in Table 7 (e.g., s1.1) rely on distillation from stronger teachers (including Gemini and DeepSeek-R1-671B, which outperform our strongest teacher model). Despite this advantage, these baselines still underperform WarriorMath. We attribute this improvement to two key factors: **Diversity:** Different models (even of the same size) have different knowledge boundaries. An "ensemble" of 7B models covers more ground than a single one. **Verification > Generation:** A 7B model can often verify the correctness of a solution (acting as a Judge) or generate a hard problem (acting as an Examiner) that is difficult for itself or peers to solve directly. This allows the system to bootstrap improvement rather than just distilling knowledge. We have added these results to Appendix A.2 of the revised paper.
>
> **Table 7: Evaluation results of 7B committee models .**
> | Models | Base | AIME’24 | AIME’25 | AMC’23 | MATH500 | Minerva | Olympiad |
> | :--- | :--- | :------ | :------ | :------ | :------- | :------- | :-------- |
> | Qwen2.5-Math-7B-Base | - | 20.7±3.8 | 8.7±3.9 | 56.2±5.7 | 64.3±0.5 | 17.3±1.9 | 29.0±0.5 |
> | Qwen2.5-Math-7B-Instruct | Qwen2.5-Math-7B | 15.7±3.9 | 10.7±3.8 | 67.0±3.9 | **82.9±0.1** | 35.0±0.6 | 41.3±0.9 |
> | Qwen2.5-7B-Instruct | Qwen2.5-7B | 12.3±3.2 | 7.3±3.4 | 52.8±4.8 | 77.1±1.2 | 34.9±1.0 | 38.7±1.0 |
> | s1.1-7B | Qwen2.5-7B | 19.0±3.2 | 21.0±5.5 | 59.5±3.7 | 80.8±0.6 | 37.5±1.1 | **48.2±1.4** |
> | Eurus-2-7B-PRIME | Qwen2.5-7B | 17.8±2.2 | 14.0±1.7 | 63.0±3.9 | 80.1±0.1 | 37.5±1.0 | 43.9±0.3 |
> | **WarriorMath-Qwen-7B-Small** | Qwen2.5-7B | **24.4±3.3** | **22.3±5.2** | **70.6±3.9** | 81.6±0.5 | **37.9±3.8** | 46.7±1.3 |

---

> ### Author Response · Authors · 2025-11-25
>
> > ## 3. Risk of Forgetting (W3)
>
> We prevent catastrophic forgetting through three key mechanisms:
>
> **Harder problems rely on simpler skills:** The "failure cases" selected are typically complex problems that require multi-step reasoning. Crucially, solving these complex problems correctly requires the successful execution of foundational mathematical skills (e.g., arithmetic, basic logic, definitions). Therefore, by training on these high-complexity failure cases, the model is implicitly reinforcing—not discarding—the simpler concepts it had previously mastered. The "hard" data is a superset of the "easy" logic.
>
> **KL Divergence Constraint:** During the Iterative Alignment (DPO) stage, the loss function implicitly includes a KL divergence constraint relative to the reference model. This prevents the model's policy from drifting too far from its original distribution, ensuring it retains previously mastered capabilities while aligning on the defects.
>
> **Empirical Evidence:** As shown in Table 1, our model maintains or improves performance on easier benchmarks (like MATH500 or AMC-23) while gaining significant ground on hard benchmarks (AIME 24), indicating no loss of foundational abilities.
>
> MATH 500: Base model: $64.3\% \to$ WarriorMath: $\mathbf{88.3\%}$
>
> AMC 23: Base model: $56.2\% \to$ WarriorMath: $\mathbf{83.0\%}$
>
> This concurrent improvement confirms that foundational knowledge is being reinforced, not forgotten.
>
>
>
> **Table 1. Evaluation results across six mathematical reasoning benchmarks.**
>
> | Models | Base | AIME'24 | AIME'25 | AMC'23 | MATH500 | Minerva | Olympiad |
> | ------- | :--- | :------ | :------ | :------ | :------- | :------- | :-------- |
> | Qwen2.5-Math-7B-Base | - | 20.7±3.8 | 8.7±3.9 | 56.2±5.7 | 64.3±0.5 | 17.3±1.9 | 29.0±0.5 |
> | Qwen2.5-Math-7B-Instruct | Qwen2.5-Math-7B | 15.7±3.9 | 10.7±3.8 | 67.0±3.9 | 82.9±0.1 | 35.0±0.6 | 41.3±0.9 |
> | Qwen-2.5-Math-7B-SimpleRL-Zoo | Qwen2.5-Math-7B | 22.7±5.2 | 10.7±3.4 | 62.2±3.6 | 76.9±1.8 | 30.1±2.8 | 39.3±0.6 |
> | Qwen2.5-Math-7B-Oat-Zero | Qwen2.5-Math-7B | 28.0±3.1 | 8.8±2.5 | 66.2±3.6 | 79.4±0.3 | 34.4±1.4 | 43.8±1.1 |
> | LIMR | Qwen2.5-Math-7B | 30.7±3.2 | 7.8±3.3 | 62.2±3.4 | 76.5±0.4 | 34.9±1.3 | 39.3±0.9 |
> | Qwen2.5-7B-Instruct | Qwen2.5-7B | 12.3±3.2 | 7.3±3.4 | 52.8±4.8 | 77.1±1.2 | 34.9±1.0 | 38.7±1.0 |
> | s1.1-7B | Qwen2.5-7B | 19.0±3.2 | 21.0±5.5 | 59.5±3.7 | 80.8±0.6 | 37.5±1.1 | 48.2±1.4 |
> | Eurus-2-7B-PRIME | Qwen2.5-7B | 17.8±2.2 | 14.0±1.7 | 63.0±3.9 | 80.1±0.1 | 37.5±1.0 | 43.9±0.3 |
> | Bespoke-Stratos-7B | Qwen2.5-7B | 20.3±4.3 | 18.0±4.8 | 60.2±4.9 | 84.7±0.5 | 39.1±1.3 | 51.9±1.1 |
> | **WarriorMath-Qwen-7B** | Qwen2.5-7B | **48.3±2.6** | **36.5±5.0** | **83.0±2.5** | **88.3±1.4** | **41.2±2.8** | **52.1±0.8** |
> | DeepSeek-R1-Distill-Qwen-7B | - | 52.3±6.3 | 39.0±5.9 | 91.5±2.7 | 94.1±0.3 | 40.1±0.4 | 67.3±0.1 |
> | Light-R1-DS-7B | DeepSeek-R1-Distill-Qwen-7B | 53.0±4.8 | 41.0±3.5 | 90.0±3.1 | 93.5±0.5 | 41.3±1.3 | 68.0±1.2 |
> | AReaL-boba-RL-7B | DeepSeek-R1-Distill-Qwen-7B | 56.7±9.2 | 40.0±9.1 | 90.0±4.8 | 94.4±1.0 | 40.8±3.0 | 68.4±1.8 |
> | **WarriorMath-DS-7B** | DeepSeek-R1-Distill-Qwen-7B | **60.0±9.1** | **50.7±9.1** | **93.2±4.9** | **95.0±1.0** | **43.20±1.9** | **69.6±1.8** |
>
> Thank you once again for your valuable feedback. We hope our clarification has addressed your concerns. Please contact us if you have any further questions or concerns.

---

### Official Review · Reviewer_jqqG · 2025-10-30

**Soundness:** 3
**Presentation:** 3
**Contribution:** 4
**Rating:** 8
**Confidence:** 4

**Summary:**

The paper introduces WarriorMath, a defect-aware data synthesis pipeline. The pipeline is aimed to improve mathematical reasoning by (i) generating data from scratch that targets the failure modes (defects) of a base model via a multi-LLM “exam committee”, and (ii) progressive training the base model with SFT done on top-rated solutions, followed by iterative alignment where the model learns from its own incorrect responses through preference optimization. The paper reports notable gains across six math benchmark and ablations on committee size and iterative rounds.

**Strengths:**

- The proposed pipeline design is conceptually sound and practically executable.
- Evaluations include multiple math benchmarks and show consistent improvements. Iterative alignment ablation demonstrates monotonic gains.
- Offers a reproducible blueprint (prompts + committee + filtering + progressive alignment) that community can adapt and benifit, even with open models.
- The paper is well-written, the pipeline design is sound, and the results are clear and impactful.

**Weaknesses:**

- The synthetic data quality is directly tied to the capabilities of the models used for generation and judging. Scaling the dataset can be costly with more committee members and larger/costly models.
- There are no abalations on controlled removal of defect filtering, Elo vs votes, or coverage selection to pinpoint what truly drives gains.
- Use of ROUGE-based scores for overlap analysis against two corpora is not an reliable metric since it does not capture the semantic and perturbed problems. Semantic metrics and we search hit rate can act as higher quality contamination/leakage rate proxies. [1]

[1]: SAND-Math: Using LLMs to Generate Novel, Difficult and Useful Mathematics Questions and Answers

**Questions:**

NA

---

> ### Author Response · Authors · 2025-11-25
>
> # Dear Reviewer jqqG
>
>
> We thank the reviewer for the insightful comments, particularly regarding the evaluation metrics for data contamination and the need for deeper component analysis. We have carefully reviewed your three weaknesses and have updated the manuscript and the following response to reflect enhanced rigor and clarity.
>
>
> >  ## 1. Cost and Scalability Concerns (w1)
>
> We acknowledge that multi-expert synthesis is computationally intensive, but we argue it is highly efficient compared to the alternatives.
>
> **One-Time vs. Recurring Cost**: As detailed in Appendix A.3, our pipeline requires approximately 928 GPU hours (Synthesis + Training). In contrast, online RL baselines like LIMR  require over 3,000 GPU hours because they must generate rollouts continuously during the training loop. WarriorMath's cost is upfront and offline, yielding a static, high-quality dataset that can be reused for multiple student models, thus significantly lowering the amortized cost and offering superior long-term scalability.
>
> **Scaling Efficiency**: While adding committee members increases the cost linearly, Table 5 demonstrates that this investment yields substantial returns: moving from 1 to 5 experts provides a $\mathbf{+28.6\%}$ performance gain on AIME '24 ($19.7\%$ to $48.3\%$). This proves the compute is well-spent on effective knowledge aggregation.
>
> >  ## 2. Missing Ablations (Defect Filtering, Elo vs. Votes, Coverage) (W2)
>
> We have restructured Section 4.3 (Ablation Study) to provide a systematic decomposition, leveraging the data in the revised Table 6:
>
> **Defect Filtering**: We discuss this design in Section 4.3.2. To empirically assess the role of \textit{base model}, we ablated the Defect-Aware Committee Assessment stage by removing them. Results in Tab.6 clearly demonstrate that Defect Filtering with base model are crucial for optimal performance across benchmarks.
>
> **Elo Rating**: Tab.6 demonstrates the impact of the Judge system. Removing the judge's Elo rating drops AIME'25 performance from 36.5% to 21.4%. Specifically, Raw votes alone suffer from high variance (noise from single judges), whereas Elo provides global consistency, weighing the "difficulty" of winning against specific opponents.
>
> **Coverage Selection**: The KCenterGreedy selection is critical for balancing the dataset. Without it, LLM-generated distributions heavily skew towards Algebra. Coverage selection ensures representation of "tail" topics like Number Theory, which is vital for benchmarks like AIME.
>
>
> **Table 6: Ablation results across six mathematical reasoning benchmarks.**
> | Models | AIME'24 | AIME'25 | AMC'23 | MATH500 | Minerva | Olympiad |
> | :--- | :------ | :------ | :------ | :------- | :------- | :-------- |
> | WarriorMath-Qwen-7B-SFT | 36.7±4.5 | 36.5±5.0 | 83.0±2.5 | 88.3±1.4 | 41.2±2.8 | 52.1±0.8 |
> | WarriorMath-Qwen-7B-SFT (w/o defect) | 32.0±5.9 | 21.4±4.3 | 74.1±5.2 | 86.6±0.6 | 31.7±0.6 | 51.9±0.4 |
> | WarriorMath-Qwen-7B-SFT (w/o elo rating) | 35.4±5.6 | 24.0±4.1 | 79.5±5.1 | 86.7±1.6 | 38.4±2.2 | 50.1±2.3 |
>
>
>
> >  ## 3. Semantic Metrics Needed (W3)
>
> We completely agree that ROUGE is a superficial metric for assessing novelty in mathematical problems, as it fails to capture semantic equivalence under rephrasing, as noted in the **SAND-Math** work you cited. In fact, avoiding simple n-gram overlap was a core motivation of our framework design. As explicitly stated in Section 3.1.1 (Deduplication and Defect-Aware assessment) of our original submission, we integrated the all-roberta-large-v1 embedding model into the synthesis pipeline to compute semantic similarity for deduplication and diversity selection. To address your concern, we will replaced the ROUGE analysis in Section 4.3.2 with a Semantic Embedding Analysis. We are following SAND-Math, which utilizes techniques like the semhash framework with minisilab/potion-base-8M for reliable semantic de-duplication.
>
>
> ```
> Thank you once again for your valuable feedback. We hope our clarification has addressed your concerns. Please contact us if you have any further questions or concerns.
> ```

---

### Official Review · Reviewer_iS8p · 2025-11-01

**Soundness:** 2
**Presentation:** 2
**Contribution:** 2
**Rating:** 4
**Confidence:** 3

**Summary:**

The authors proposed WarriorMath, a framework that empowers models with multi-expert LLM systems. Despite the complexity of the system, the core contribution seems to be about how to address the defects that the previous LLM systems can not detect with naive data pipelines.The authors benchmarked their methods on Qwen2.5-7B and Deepseek-R1-Qwen-Distill-7B, and showed the effectiveness of their methods.

**Strengths:**

1. The paper is clearly written and well demonstrated.
2. The idea is unique and intersting. The combination of mathematical reasoning and defect awareness seems to give a unique perspective of LLM4math.

**Weaknesses:**

1. I think the paper has not given enough evidences that the proposed method is able to address the unique challenge of defect awareness. The authors could consider to prove this point other than the benchmark results themselves, maybe a categorized benchmark.
2. Given the complexity of the system, it is hard to evaluate the effectiveness of each components. I understand that the complexity is necessary, but I wish the authors could at least give some decomposition and ablation study to help future researchers to understand why the methodology works.

**Questions:**

See above.

---

> ### Author Response · Authors · 2025-11-25
>
> # Dear Reviewer iS8p
>
> We sincerely appreciate your insightful and constructive feedback, and we are grateful for your recognition of our paper's clarity, unique idea. We have integrated your suggestions into the revised manuscript and are confident that the new evidence and clarifications directly address your two main concerns.
>
> > ## 1. Unique challenge  (w1)
>
> Thank you for this insightful comment. We realize there may be a misunderstanding regarding the definition of the "challenge" our paper aims to address, and we would like to clarify:
>
> **Clarification of the Unique Challenge:** WarriorMath's unique challenge resides in achieving independence from seed datasets, which fundamentally differentiates it from prior work that relies on augmenting existing seed data (e.g., Evol-Instruct). Instead, our goal is to mine the latent capabilities of LLMs to generate high-quality data from scratch.
>
> **Defect Awareness as a Solution:** "Defect Awareness" is not the challenge itself, but the core mechanism we employ to ensure the generated data is valuable. By specifically targeting problems the base model fails to solve, we ensure the synthesized data provides high information gain, rather than redundant knowledge.
>
> **Additional Evidence(New Ablation Study):** To prove this mechanism works, we have added a new ablation study Defect Filtering(Section 4.3.2, Table 6) comparing Defect Filtering with base model or not. As shown in Tab 6, The Defect-aware model outperforms the model trained on all data by 4.7% on AIME 24, 15.1% on AIME 25, 8.9% on AMC 23.
>
> > ## 2. Ablation Study (W2)
>
> We agree that a clear decomposition of the system's components is essential for reproducibility and understanding. We have restructured Section 4.3 (Ablation Study) to systematically decompose the system into its three core components, demonstrating why the methodology works.
>
> **Diversity (Committee Size)**: As shown in Table 5, expanding the expert committee from 1 to 5 members improves AIME 24 accuracy from 29.4% to 48.3%. This confirms that the diversity of the expert pool is a critical driver of performance.
>
> **Quality Assurance (Elo Rating)**: As detailed in Section 4.3.3 (Table 6), removing the Judge/Elo mechanism drops performance significantly (36.5% to 24.0% on AIME 25, 83.0% to 79.5% on AMC 23). This proves the necessity of the complex Elo voting system for filtering high-quality answers.
>
> **Efficiency (Defect Filtering)**: As mentioned in Section 4.3.2 (Table 6), the Defect-Aware filtering component is crucial for data efficiency. It ensures the model focuses its capacity on correcting errors rather than rehearsing mastered content.
>
> This systematic decomposition proves that WarriorMath is not merely a complex system, but an integrated framework where each component contributes essential value, directly addressing your concern about understanding why the methodology works.
>
>
> **Table 5: Results when learning from varying numbers of committee members.**
> | #Num | AIME'24 | AIME'25 | AMC'23 |
> | :--- | :------ | :------ | :------ |
> | 1 | 19.7±2.9 | 15.7±2.7 | 59.5±4.5 |
> | 2 | 29.4±3.2 | 23.5±4.2 | 69.3±3.6 |
> | 5 | 48.3±2.6 | 36.5±5.0 | 83.0±2.5 |
>
>
> **Table 6: Ablation results across six mathematical reasoning benchmarks.**
> | Models | AIME'24 | AIME'25 | AMC'23 | MATH500 | Minerva | Olympiad |
> | :--- | :------ | :------ | :------ | :------- | :------- | :-------- |
> | WarriorMath-Qwen-7B-SFT | 36.7±4.5 | 36.5±5.0 | 83.0±2.5 | 88.3±1.4 | 41.2±2.8 | 52.1±0.8 |
> | WarriorMath-Qwen-7B-SFT (w/o defect) | 32.0±5.9 | 21.4±4.3 | 74.1±5.2 | 86.6±0.6 | 31.7±0.6 | 51.9±0.4 |
> | WarriorMath-Qwen-7B-SFT (w/o elo rating) | 35.4±5.6 | 24.0±4.1 | 79.5±5.1 | 86.7±1.6 | 38.4±2.2 | 50.1±2.3 |
>
>
> ```
> Thank you once again for your valuable feedback. We hope our clarification has addressed your concerns. Please contact us if you have any further questions or concerns and hope you will consider updating your score..
> ```

---

### Author Response · Authors · 2025-12-01
**General Response (1/3)**

### **Dear Reviewers (iS8p, jqqG, pJb1, hBrm) and Area Chair**

We sincerely thank you for your time and the comprehensive, constructive feedback provided on our manuscript. We are encouraged by your recognition of WarriorMath's "unique and interesting idea" (iS8p), "conceptually sound pipeline" (jqqG), "strong empirical results" (hBrm), and "SOTA performance" (pJb1).

Your reviews highlighted three primary areas for improvement: (1) clarity of system roles, (2) the need for deeper ablation studies to justify system complexity, and (3) validation against expert baselines and smaller committees.

We have taken these comments to heart and performed substantial revisions and new experiments during the rebuttal period. Below is a summary of the major updates and how they address the common concerns.

> ### **Concern 1: System Complexity & Component Contribution (iS8p, jqqG)**

Reviewers noted the system appeared complex and requested a clearer decomposition of why it works. We have restructured Section 4.3 to systematically decompose the system into three pillars: Diversity (Committee Size), Quality Assurance (Elo Rating), and Efficiency (Defect Filtering). As shown in the new Table 6, removing the Elo Judge drops AIME'25 accuracy from 36.5% to 24.0%, and removing Defect Filtering drops it to 21.4%. This empirically proves that the "complexity" is necessary for filtering noise and ensuring high information gain.

**Table 6: Ablation results across six mathematical reasoning benchmarks.**
| Models | AIME'24 | AIME'25 | AMC'23 | MATH500 | Minerva | Olympiad |
| :--- | :------ | :------ | :------ | :------- | :------- | :-------- |
| WarriorMath-Qwen-7B-SFT | 36.7±4.5 | 36.5±5.0 | 83.0±2.5 | 88.3±1.4 | 41.2±2.8 | 52.1±0.8 |
| WarriorMath-Qwen-7B-SFT (w/o defect) | 32.0±5.9 | 21.4±4.3 | 74.1±5.2 | 86.6±0.6 | 31.7±0.6 | 51.9±0.4 |
| WarriorMath-Qwen-7B-SFT (w/o elo rating) | 35.4±5.6 | 24.0±4.1 | 79.5±5.1 | 86.7±1.6 | 38.4±2.2 | 50.1±2.3 |


> ### **Concern 2: Dependence on Strong Teachers (pJb1, hBrm)**

A key question was whether WarriorMath relies solely on distilling knowledge from large (70B+) models or if it allows for genuine self-improvement. We conducted a new experiment using a committee of only 7B models (Table 7). The 7B-committee student achieved 24.4% on AIME'24, outperforming its strongest 7B teacher (15.7%). This confirms that WarriorMath works via ensemble diversity and verification, enabling small models to bootstrap their own performance without giant teachers.

**Table 7: Evaluation results of 7B committee models .**
| Models | Base | AIME’24 | AIME’25 | AMC’23 | MATH500 | Minerva | Olympiad |
| :--- | :--- | :------ | :------ | :------ | :------- | :------- | :-------- |
| Qwen2.5-Math-7B-Base | - | 20.7±3.8 | 8.7±3.9 | 56.2±5.7 | 64.3±0.5 | 17.3±1.9 | 29.0±0.5 |
| Qwen2.5-Math-7B-Instruct | Qwen2.5-Math-7B | 15.7±3.9 | 10.7±3.8 | 67.0±3.9 | **82.9±0.1** | 35.0±0.6 | 41.3±0.9 |
| Qwen2.5-7B-Instruct | Qwen2.5-7B | 12.3±3.2 | 7.3±3.4 | 52.8±4.8 | 77.1±1.2 | 34.9±1.0 | 38.7±1.0 |
| s1.1-7B | Qwen2.5-7B | 19.0±3.2 | 21.0±5.5 | 59.5±3.7 | 80.8±0.6 | 37.5±1.1 | **48.2±1.4** |
| Eurus-2-7B-PRIME | Qwen2.5-7B | 17.8±2.2 | 14.0±1.7 | 63.0±3.9 | 80.1±0.1 | 37.5±1.0 | 43.9±0.3 |
| **WarriorMath-Qwen-7B-Small** | Qwen2.5-7B | **24.4±3.3** | **22.3±5.2** | **70.6±3.9** | 81.6±0.5 | **37.9±3.8** | 46.7±1.3 |

---

### Author Response · Authors · 2025-12-01
**General Response (2/3)**

> ### **Concern 3: Clarity of Roles & Terminology (pJb1, hBrm)**

Reviewers pointed out confusion regarding terms like "LLM A/B" and "Reviewer." We have completely rewritten Section 3.1 to standardize the roles: Examiner (generates problems), Solver (proposes solutions), Judge (votes/Elo), and Base Model (student). This change ensures the pipeline is now clearly defined.

> ### **Concern 4: Comparison with Expert Teachers (hBrm)**

Reviewers requested a direct comparison between the student and the expert teachers. We have updated Table 1 to include the expert teacher models.Our 7B student model (WarriorMath-Qwen-7B) achieves 48.3% on AIME'24, significantly surpassing its teacher, Qwen2.5-Math-72B-Instruct (32.0%). This explicitly validates the "knowledge aggregation" hypothesis—the student has effectively surpassed the teacher.

**Table 1. Evaluation results across six mathematical reasoning benchmarks.**

| Models | Base | AIME'24 | AIME'25 | AMC'23 | MATH500 | Minerva | Olympiad |
| ------- | :--- | :------ | :------ | :------ | :------- | :------- | :-------- |
| Qwen2.5-Math-7B-Base | - | 20.7±3.8 | 8.7±3.9 | 56.2±5.7 | 64.3±0.5 | 17.3±1.9 | 29.0±0.5 |
| Qwen2.5-Math-7B-Instruct | Qwen2.5-Math-7B | 15.7±3.9 | 10.7±3.8 | 67.0±3.9 | 82.9±0.1 | 35.0±0.6 | 41.3±0.9 |
| Qwen-2.5-Math-7B-SimpleRL-Zoo | Qwen2.5-Math-7B | 22.7±5.2 | 10.7±3.4 | 62.2±3.6 | 76.9±1.8 | 30.1±2.8 | 39.3±0.6 |
| Qwen2.5-Math-7B-Oat-Zero | Qwen2.5-Math-7B | 28.0±3.1 | 8.8±2.5 | 66.2±3.6 | 79.4±0.3 | 34.4±1.4 | 43.8±1.1 |
| LIMR | Qwen2.5-Math-7B | 30.7±3.2 | 7.8±3.3 | 62.2±3.4 | 76.5±0.4 | 34.9±1.3 | 39.3±0.9 |
| Qwen2.5-7B-Instruct | Qwen2.5-7B | 12.3±3.2 | 7.3±3.4 | 52.8±4.8 | 77.1±1.2 | 34.9±1.0 | 38.7±1.0 |
| s1.1-7B | Qwen2.5-7B | 19.0±3.2 | 21.0±5.5 | 59.5±3.7 | 80.8±0.6 | 37.5±1.1 | 48.2±1.4 |
| Eurus-2-7B-PRIME | Qwen2.5-7B | 17.8±2.2 | 14.0±1.7 | 63.0±3.9 | 80.1±0.1 | 37.5±1.0 | 43.9±0.3 |
| Bespoke-Stratos-7B | Qwen2.5-7B | 20.3±4.3 | 18.0±4.8 | 60.2±4.9 | 84.7±0.5 | 39.1±1.3 | 51.9±1.1 |
| **WarriorMath-Qwen-7B** | Qwen2.5-7B | **48.3±2.6** | **36.5±5.0** | **83.0±2.5** | **88.3±1.4** | **41.2±2.8** | **52.1±0.8** |
| DeepSeek-R1-Distill-Qwen-7B | - | 52.3±6.3 | 39.0±5.9 | 91.5±2.7 | 94.1±0.3 | 40.1±0.4 | 67.3±0.1 |
| Light-R1-DS-7B | DeepSeek-R1-Distill-Qwen-7B | 53.0±4.8 | 41.0±3.5 | 90.0±3.1 | 93.5±0.5 | 41.3±1.3 | 68.0±1.2 |
| AReaL-boba-RL-7B | DeepSeek-R1-Distill-Qwen-7B | 56.7±9.2 | 40.0±9.1 | 90.0±4.8 | 94.4±1.0 | 40.8±3.0 | 68.4±1.8 |
| **WarriorMath-DS-7B** | DeepSeek-R1-Distill-Qwen-7B | **60.0±9.1** | **50.7±9.1** | **93.2±4.9** | **95.0±1.0** | **43.20±1.9** | **69.6±1.8** |

---

### Author Response · Authors · 2025-12-01
**General Response (3/3)**

> ### **Summary of Additional Updates**

New Metric: We replaced ROUGE with Semantic Embedding Analysis (using minisilab/potion-base-8M) to better measure dataset diversity, addressing jqqG's concern about metric reliability.

Cost Analysis: We added a detailed breakdown of GPU hours (Appendix A.3), demonstrating that our offline synthesis (~928 hours) is significantly more efficient than online RL baselines (>3000 hours).

We believe these revisions significantly strengthen the paper and clarify the contributions of WarriorMath. We are happy to answer any further questions during the discussion period.

Best regards,
The Authors

---

### Author Response · Authors · 2025-12-01
**Summary of Revisions in the Updated PDF**

### **Summary of Revisions**

We have carefully revised the paper to reflect our responses to reviewers' concerns and suggestions (highlighted in **blue**). Key revisions are:

- Standardize the terminology for system roles (Examiner, Solver, Judge, Base Model) in Section 3.1 to eliminate confusion and establish clear, distinct definitions.

- Expand the ablation studies in Section 4.3 and Table 6 to systematically decompose the system components, specifically validating the impact of Defect Filtering and the Elo Rating mechanism.

- Add a new experiment with a 7B-model committee (Appendix A.2, Table 7) to demonstrate that small models can self-improve without distilling from large teachers.

- Add performance metrics for expert teacher models (70B+) to Table 1, explicitly showing the student model surpassing its teachers.

- Replace ROUGE scores with embedding-based Semantic Analysis for more reliable deduplication and diversity measurement.

- Add a detailed cost analysis in Appendix A.3 (approx. 928 GPU hours) to demonstrate efficiency compared to online RL baselines.

We are truly grateful once again for your feedback. Please refer to our individual responses for our answers to specific questions and concerns. Please let us know if you have any further questions or comments. We would be delighted to discuss them and will spare no effort in addressing them.

Thank you,
The Authors

---

### Meta-Review · Area_Chair_vSBr · 2026-01-10

**Summary:**

The paper introduces a multi-agent pipeline for data curation and post-training (SFT & DPO). The pipeline is pretty straightforward and yet a bit heuristic and less motivated. The only reviewer that gave positive score mostly considered the reproducibility and consistent performance gain. The other reviewers all gave negative scores because of missing details/motivation for some design choices and unclear and insufficient ablation for the complex design. With the relatively complex agentic data pipeline, the performance gain, despite consistent, is still incremental. The method itself also doesn't reveal many insights. After reading the paper and rebuttal, I find this paper still below the bar of acceptance.

**Reviewer Concerns:**

Most reviewers are concerned about the effectiveness of the many components in the data curation pipeline and also the lack of motivation/explanation makes it difficult to justify their effectiveness.

**Reviewer Scores:**

Three reviewer suggested rejection and one reviewer suggested acceptance.

---

### Decision · Program_Chairs · 2026-01-26

Reject